# The 1600 Huaynaputina Eruption as Possible Trigger for Persistent Cooling in the North Atlantic Region

Sam White[1], Eduardo Moreno-Chamarro[2], Davide Zanchettin[3], Heli Huhtamaa[4,5], Dagomar Degroot[6], Markus Stoffel[7,8,9], Christophe Corona[7,10]

[1] Department of History, Ohio State University, Columbus, Ohio, 43210, United States
[2] Barcelona Supercomputing Center (BSC), Barcelona, Spain
[3] Department of Environmental Sciences, Informatics and Statistics, University Ca' Foscari of Venice, Mestre, 30172, Italy
[4] Institute of History, University of Bern, 3012 Bern, Switzerland
[5] Oeschger Centre for Climate Change Research, University of Bern, 3012 Bern, Switzerland
[6] Department of History, Georgetown University, Washington, DC, 3700, United States
[7] Climatic Change Impacts and Risks in the Anthropocene (C-CIA), Institute for Environmental Sciences, University of Geneva, Geneva, 1205, Switzerland
[8] Department for Earth Sciences, University of Geneva, Geneva, 1205, Switzerland
[9] Department F.-A. Forel for Aquatic and Environmental Sciences, University of Geneva, Geneva, 1205, Switzerland
[10] CNRS, Geolab, University of Clermont Auvergne, Clermont-Ferrand, 63000, France

*Correspondence to*: Sam White (white.2426@osu.edu)

**Abstract.** Paleoclimate reconstructions have identified a period of exceptional summer and winter cooling in the North Atlantic region following the eruption of the tropical volcano Huaynaputina (Peru) in 1600 CE. A previous study based on numerical climate simulations has indicated a potential mechanism for the persistent cooling in a slowdown of the North Atlantic subpolar gyre (SPG) and consequent ocean-atmosphere feedbacks. To examine whether this mechanism could have been triggered by the Huaynaputina eruption, this study compares the same simulations used in the previous study both with and without volcanic forcing and this SPG shift to reconstructions from annual proxies in natural archives and historical written records as well as contemporary historical observations of relevant climate and environmental conditions. These reconstructions and observations demonstrate patterns of cooling and sea ice expansion consistent with, but not indicative of, an eruption trigger for the proposed SPG slowdown mechanism. The results point to possible improvements in future model-data comparison studies utilizing historical written records. Moreover, we consider historical societal impacts and adaptations associated with the reconstructed climatic and environmental anomalies.

## 1 Introduction

This article draws on high-resolution climate proxies and direct historical observations to examine whether the Huaynaputina eruption in 1600 CE could have triggered a mechanism identified in previous modeling studies (Moreno-Chamarro et al., 2017a, 2017b) for persistent summer and winter cooling in the North Atlantic region. Huaynaputina is a stratovolcano in southern Peru (4,850m asl; 16.61°S, 70.85°W) and belongs to the Central Volcanic Zone of the Andean Volcanic Belt originating in the subduction of the oceanic Nazca plate beneath the continental South American plate (Global Volcanism Project, 2021). The 1600 Huaynaputina eruption is rated 6 on the volcanic explosivity index (VEI); it was the largest eruption in the Andes in historical times and the largest source of dust in the annually resolved Quelccaya ice-core record (Thompson et al., 2013). According to contemporary written accounts, tremors began to be felt in the region in mid-February 1600. A large Plinian eruption took place on February 19, followed by a low, heavy ashfall that blackened the sky for tens of kilometers around. Pyroclastic flows continued through February 26, and a phase of vulcanian eruption continued to eject pumice and ash through approximately the beginning of March, with a total ejection of ~13-14 km³ of tephra reaching up to 400 km from the volcano (Jara et al., 2000; Adams et al., 2001; Thouret et al., 2002; Prival et al., 2019). Pyroclastic flows, lahars, meltwater flooding, and earthquakes caused widespread damage and loss of life in the surrounding region (De Silva et al., 2000).

The first years of the 17<sup>th</sup> century stand out as some of the coldest of the past two millennia in multiple Northern Hemisphere (NH) summer temperature reconstructions, particularly those based on tree-ring density (Schneider et al., 2015; Stoffel et al., 2015; Guillet et al., 2017). Several reconstructions rank 1601 as the coldest summer and/or 1600-09 as the decade with the coldest NH summer temperatures in at least the past 420 years (D'Arrigo et al., 2006; D'Arrigo et al., 2009). Moreover, 1601 stands out as an extreme cold and/or dry summer in numerous regional climate reconstructions around the North Atlantic, including those for Quebec (Gennaretti et al., 2014) and Scandinavia (McCarroll et al., 2013). This abrupt and exceptional cooling has been attributed to the direct radiative response from enhancement of the stratospheric aerosol layers by the Huaynaputina eruption (De Silva and Zielinksi, 1998; Briffa et al., 1998; Verosub and Lippman, 2008). Previous studies have linked this eruption to a volcanic dust veil and reported dimming of sunlight (Lamb, 1970). For example, a Russian chronicle records persistent dark skies in spring 1601 (Akiander, 1849), pointing to the persistence of aerosols in the stratosphere more than a year after the eruption. Sulfate depositions dated to ca. 1600 in ice cores at both poles and the tropics have also been attributed to the Huaynaputina eruption (De Silva et al., 1998; Thompson et al., 2013).

Despite the large body of paleoclimatic and historical sources, the connection between the Huaynaputina eruption and concomitant climatic anomalies poses unresolved research issues. Ice core records with robust sampling indicate that the Huaynaputina eruption injected fewer sulfates into the stratosphere than other Little Ice Age (LIA) eruptions with a comparable cooling effect, such as the 1453 Kuwae or the 1815 Tambora eruptions, and that these sulfates were asymmetrically distributed between the Northern and Southern Hemispheres (Sigl et al., 2015; Toohey and Sigl, 2017). European historical records and temperature reconstructions utilizing written records of weather observations and climate proxies—the "archives of societies" (Brönnimann et al., 2018)—indicate that there was more exceptional winter cooling than summer cooling in the years following the eruption, particularly in Central and Northern Europe (Luterbacher et al., 2004; Pfister et al., 2018). Most importantly, the observed summer and especially winter cooling persisted longer than would be expected from the direct radiative forcing by transient stratospheric volcanic aerosols alone (Stoffel et al., 2015; Toohey et al., 2019).

Previous work in numerical climate modeling found a potential mechanism for persistent cooling, particularly in and around the North Atlantic starting ca. 1600 (Moreno-Chamarro et al., 2017a). In this work, simulations with the Max Planck Institute -Earth System Model in its CMIP5 configuration for paleoclimate studies (MPI-ESM-P) revealed a mechanism featuring sea-ice expansion and reduced ocean heat losses in the Nordic and Barents seas driven by a reduction in the northward heat transport by the North Atlantic subpolar gyre (SPG). In this mechanism, feedbacks between North Atlantic cooling and sea-ice buildup, on the one hand, and sub-Arctic atmospheric pressure anomalies, on the other, lead to an increased frequency of atmospheric blocking conditions over Europe, in turn driving extreme winter cooling over the continent and particularly over Scandinavia and Central Europe. This mechanism does not entail significant changes in the dominant mode of large-scale atmospheric variability over the North Atlantic, namely the North Atlantic Oscillation (NAO), which is often involved in post-eruption interannual and decadal climate variability (e.g., Zanchettin et al., 2013). Previous studies have demonstrated agreement between simulations with the SPG slowdown and paleoclimate reconstructions regarding the long-term seasonally asymmetric cooling in the North Atlantic region (Moreno-Chamarro, 2017b).

However, ensemble sensitivity simulations with MPI-ESM-P and imposed forcing but different initial conditions have called into question whether volcanic forcing would have been necessary to trigger such an SPG shift. In particular, they have revealed substantial uncertainties about the initial anomaly in Arctic freshwater export that triggers the SPG-centered feedback loop (Moreno-Chamarro et al., 2017b). Previous studies have drawn on proxies in natural archives at mostly decadal or lower resolution to determine the magnitude and timing of shifts in ocean sea-surface temperature and sea-ice extent characteristic of the SPG slowdown (Moreno-Chamarro et al., 2017a, 2017b). Some historical climatology evidence indicates cooling in Europe and the North Atlantic prior to the 1600 eruption. Winter temperature reconstructions of Central Europe based on contemporary written observations indicate an onset of colder winters by the 1590s (Dobrovolný et al., 2010; Pfister and Wanner, 2021). Furthermore, historic fishing records indicate an expanded herring catch at the southern end of the herring

range during the 1590s, which would also be consistent with cooling North Atlantic temperatures prior to 1600, although this shift may have alternative explanations (Alheit and Hagen, 1997; Holm et al., 2019). Therefore, further higher-resolution reconstructions and examination of historical observations are required to determine whether such an SPG shift was triggered by the Huaynaputina eruption.

The question of whether the Huaynaputina eruption triggered such an SPG slowdown mechanism and thus persistent cooling in the North Atlantic region has implications for human history as well. Previous research associates climatic anomalies— particularly cool, wet summers—with harvest failures, price spikes, excess mortality, and civil unrest across Europe during the 1590s and early 17[th] century (Clark, 1985; Parker, 2013; Parker, 2018; Degroot, 2018). These impacts included severe subsistence crises in Scandinavia (Utterström, 1955; Huhtamaa and Helama, 2017) and Russia (Dunning, 2001) during

exceptionally cold years. By constraining the timing of summer and winter cooling and the role of the Huaynaputina eruption therein, we can help determine the relative roles of climatic versus non-climatic causes, volcanic forcing versus intrinsic variability of the climate system, and short-term anomalies versus longer-term climatic changes in those societal impacts. Moreover, by analyzing climatic changes in both summer and winter at high temporal and spatial resolution, as well as environmental changes such as sea ice duration and extent, this study may help specify which activities and institutions

were vulnerable, and how people adapted to changing climatic and environmental conditions ca.1600 CE.

Therefore, this study applies new high-resolution paleoclimate and historical data to determine whether the 1600 Huaynaputina eruption triggered the previously identified SPG slowdown mechanism and persistent cooling in the early 17[th] century. It compares the MPI-ESM-P simulations used in Moreno-Chamarro et al. (2017a, 2017b) with and without volcanic forcing and SPG shift to reconstructions from annual proxies in natural archives and historical written records as well as direct historical

observations of relevant climate and environmental conditions. Our choice to limit the analysis to the previous MPI-ESM-P simulations is motivated by the fact that a multi-model investigation including additional simulations covering the early 17[th] century would have added several sources of uncertainty such as different model sensitivities, different volcanic forcings, different background climate states at the time of the eruption, and different internal variability. The only set of simulations with a close enough setup is the last millennium ensemble with the Community Earth System Model- Community Atmosphere

Model version 5 (CESM-CAM5), which includes simulations of the last millennium with different external forcing. However, it is not clear whether CESM-CAM5 shows any sensitivity in the subpolar region to the volcanic forcing (Otto-Bliesner et al., 2016), although a newer version of the model shows cooling in the North Atlantic during the LIA associated with an SPG weakening (Zhong et al., 2018).

We seek to establish whether the years following the eruption saw an abrupt shift in mean conditions of sea-ice extent and/or

winter temperature that are characteristic traits of the SPG mechanism and whether particular climatic conditions at the time of the Huaynaputina eruption determined activation (or lack thereof) of the SPG slowdown mechanism. In this way, the study demonstrates the potential for combining natural archives and written historical records to test model-derived mechanisms during the pre-instrumental period, and thus to further the integration of climate modeling, paleoclimatology, and historical climatology for a better understanding of past climate variability and its human dimensions.

**2. A Mechanism for Persistent Cooling ca. 1600 CE and Current Paleoclimate Evidence**

This section reviews the previous modeling results that established the SPG mechanism for persistent cooling and its consistency with previous paleoclimate reconstructions, as well as the challenges in determining whether or not this mechanism was triggered by the 1600 Huaynaputina eruption. Volcanic eruptions such as Huaynaputina influence climate by ejecting chemically and microphysically active gases and solid particles into the atmosphere (Robock, 2000; Timmreck, 2012). During

strong eruptions, the volcanic column penetrates the lower stratosphere, where sulfur-containing gases turn into aerosol particles that enhance the existing stratospheric aerosol layer. Volcanic aerosols then typically persist in the stratosphere for up to two years (e.g., Timmreck, 2012; Zanchettin et al., 2012). Aerosol particles influence the Earth's radiative balance both

by scattering incoming solar radiation back to space, which cools the troposphere and the surface, and by absorbing long-wave solar and terrestrial radiation, which locally warms the stratosphere.

Although these radiative effects are short-lived, volcanic eruptions may influence the decadal and longer evolution of regional climates in two ways (e.g., Zanchettin, 2017). First, close successions of small or moderate eruptions—so-called volcanic clusters—may lead to prolonged periods of surface cooling associated with the persistence of an enhanced stratospheric aerosol layer. Second, dynamical responses may initiate decadal or multidecadal feedback loops encompassing large-scale atmospheric circulation, sea ice, and oceanic thermohaline circulation (e.g., Otterå et al., 2010; Zanchettin et al., 2012).

Moreno-Chamarro et al. (2017b) have investigated such a scenario of eruption-induced persistent cooling for the period around 1600 CE. According to their paleoclimate model simulations, the late 16$^{th}$-century cluster of volcanic eruptions culminating in Huaynaputina produced a cumulative global-average negative radiative flux of about -5 Wm$^{-2}$ at the top of the atmosphere (Fig. 1a; see also Moreno-Chamarro et al., 2017b). This volcanic cluster includes the 1585 eruption of Colima, Mexico (VEI 4), a 1591 unidentified eruption, and the 1595 eruption of Nevado del Ruiz, Colombia (VEI 4) observed by contemporaries

(Thouret et al., 1990; Simón, 1892, pp. 127-129). Each of these eruptions injected sulfates into the stratosphere (Sigl et al., 2013) and had significant short-term cooling effects on NH summer temperatures (Sigl et al., 2015).

As shown by paleoclimate simulations of the last millennium, this cluster of eruptions could have contributed to the onset and persistence of LIA-like climate anomalies (Helama et al., 2021) for two or more centuries due to a sudden increase in the freshwater export from the Arctic that pushed the SPG into a stable weak state (Moreno Chamarro et al., 2017a), as illustrated

in **Figure 1**. More specifically, volcanic cooling abruptly increased Arctic sea ice and the associated freshwater export (Figure 1b), which induced freshening of the upper subpolar North Atlantic (Figure 1c). The freshening decreased upper ocean density in the Labrador Sea and thereby weakened the baroclinic zonal density gradient (Figure 1d) that controls the SPG strength (e.g., Born and Stocker, 2014). Once in a weak mode, the SPG induced long-lasting anomalies in the surrounding subpolar and polar ocean, affecting the oceanic transports of heat and salt as well as the sea ice, which helped sustain the weak SPG

mode and the associated anomalous climate state (Figure 1e). The existence of bistability in the SPG in different climate models (Born et al., 2013) supports the possibility of such a regime shift of the subpolar North Atlantic Ocean. Moreno Chamarro et al. (2017b) found consistencies at multidecadal scales between simulations with a weakened SPG and reconstructed changes in several geophysical variables of the North Atlantic after ca. 1600 CE.

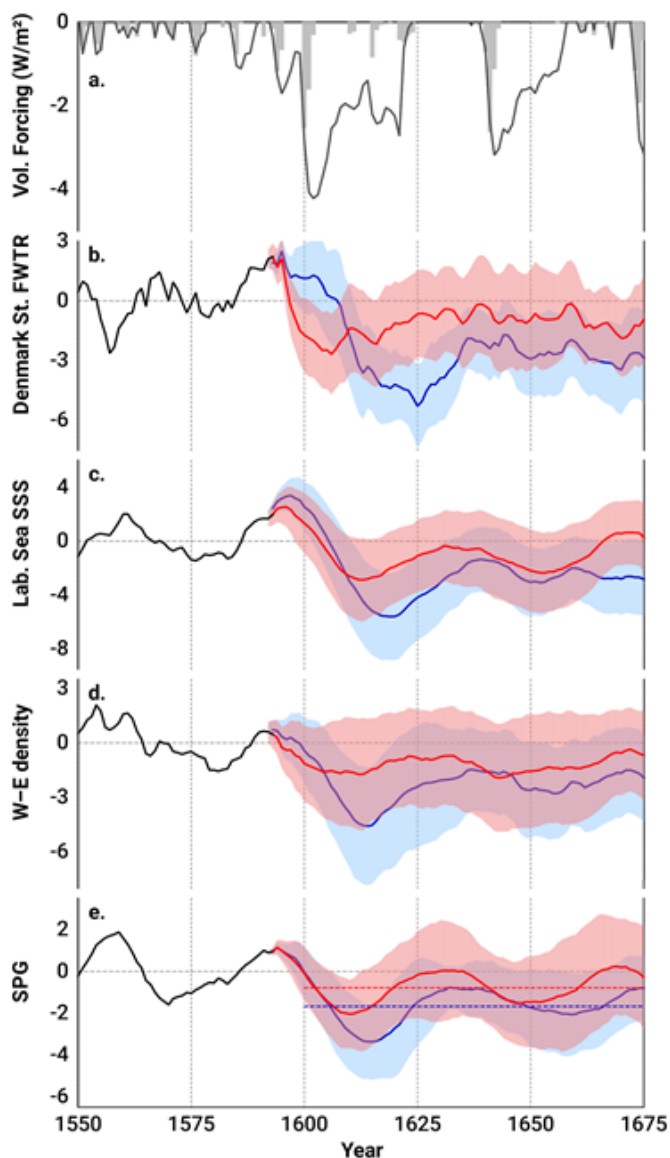

**Figure 1: Evolution of North Atlantic climate around the 1600 Huaynaputina eruption from ensemble simulations with MPI-ESM-P (after Moreno-Chamarro et al., 2017a). Blue: realizations with a subpolar gyre (SPG) shift around 1600; red: realizations with no SPG shift. Continuous lines: means of realizations; shadings: standard deviation across realizations; dashed lines: post-1600 temporal averages showing the change in regime in the SPG. From top: (a) Accumulated global top-of-the-atmosphere radiative volcanic forcing (line) and the contribution of each individual eruption (bars), both in W/m²; (b) Denmark Strait freshwater transport (FWTR), as the sum of the liquid and the sea-ice contributions (a larger negative value reflects a stronger southward export); (c) Labrador Sea sea-surface salinity (SSS); (d) West–east gradient in upper-ocean densities in the subpolar North Atlantic; and (e) SPG strength. (b–e) Time series normalized with respect to the pre-eruption period 1550–1590.**

Sensitivity simulations of the period 1593-1650 with no volcanic forcing yielded SPG shifts similar to those in the volcanically forced simulations. Hence, the study did not conclude that the late 16[th]-century volcanic cluster was necessary for the SPG shift, which was instead mainly attributed to intrinsic variability of the simulated climate system. Nevertheless, simulations with eruptions produced SPG shifts more frequently than those without: In the 10 runs of the ensemble with volcanic forcing, 6 produced the SPG shift, and 4 did not; while in the 10 runs of the ensemble without volcanic forcing 2 produced the SPG shift, and 8 did not. Therefore, even in volcanically forced simulations, the interplay between the volcanic forcing and ongoing intrinsic climate variability determined how the eruptions shaped the SPG response. This finding agrees with the general consideration that the climatic response to eruptions depends on the background climate state (Zhong et al., 2011; Zanchettin et al., 2012, 2013; Pausata et al., 2015, 2016; Gagné et al., 2017).

The dependency of the response on background conditions thus poses a barrier to attribution of the SPG shift around 1600 CE. One possible way to overcome this barrier would be identification among the ensemble sensitivity simulations of a local anomaly that preceded the enhanced Arctic freshwater release and thus determined the SPG shift. For this study, a search for

such an "initial seed" was performed in model variables such as sea-surface temperature, sea ice, and oceanic circulation in the sensitivity simulations; however, no evident initial conditions were identified. Nor did we find that the onset of the SPG shift depended on a particular state of the NAO. Neither simulations with an SPG nor those without display consistent or

anomalous high or low NAO index values in the decades before or after 1600, as shown in **Figure 2**. Furthermore, different reconstructions indicate different NAO index values during this period (Figure 2 and Hernandez et al., 2020), but neither the reconstructions nor simulations display a positive NAO anomaly after 1600 such as those identified following other large tropical eruptions (e.g., Christiansen, 2008). As recent studies indicate (Bittner et al., 2016; Coupe and Robock, 2021), a post-eruption positive NAO response with Eurasian winter warming appears to be contingent on tropospheric conditions at the time

of the eruption rather than a dynamical response to stratospheric aerosols alone. Therefore, the precise conditions for the onset of the mechanism may be related either to stochastic phenomena or to a combination of anomalous states in different climate components, or else they may be undetectable within the limits of the small ensemble size investigated by Moreno-Chamarro et al. (2017b).

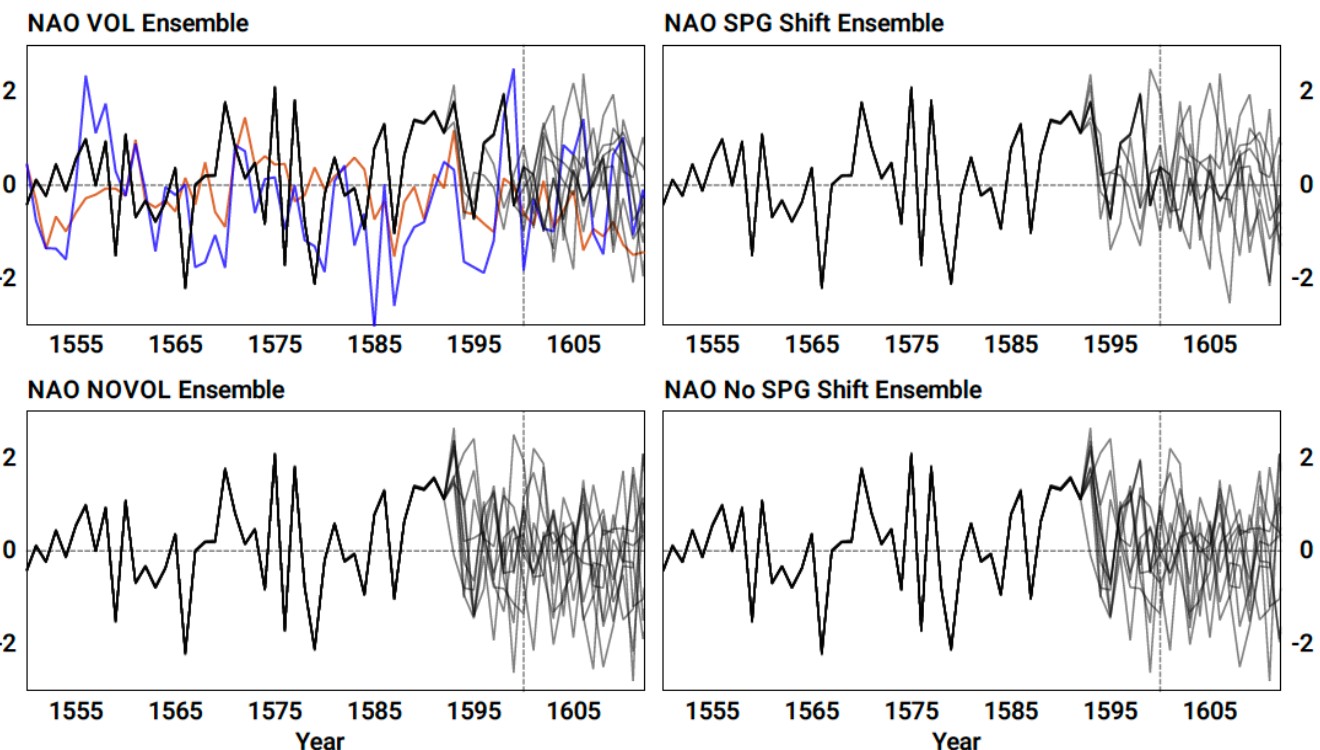

**Figure 2: North Atlantic Oscillation (NAO) time series (defined as the first principal component of winter, DJF, sea-level pressure over the North Atlantic, 20°N–90°N and 80°W–40°E) in the MPI-ESM-P ensemble with and without volcanic forcing (left) and with and without subpolar gyre (SPG) shift (right). Two NAO reconstructions are plotted from Ortega et al. (2015; red) and Michel et al. (2020; blue).**

Since an initial seed could not be identified, we have attempted a second approach to attribution in this study. The simulations with the SPG slowdown show short-term and long-term climatic and environmental changes with distinguishing traits, which may constrain the combinations of external forcing and climate state compatible with this mechanism. In 1601 and 1602, these simulations show a widespread summer cooling, especially around the Mediterranean Sea, Greenland, and the eastern North Atlantic, as shown in **Figure 3**. This short-term cooling mainly reflects the direct radiative response to the volcanic forcing,

since 6 out of the 8 simulations that produce the SPG shift are those with volcanic forcing. In the short term, simulations with an SPG shift also show minor differences in oceanic variables such as the barotropic stream function and winter sea-surface temperature in the North Atlantic. Larger differences between ensembles with and without an SPG shift over the following decades, particularly after the 1610s, as shown in **Figure 4**. The weakening in the circulation in the subpolar North Atlantic, which represents the SPG slowdown, is accompanied by cooling and expanded sea ice in the Nordic Seas, in response to

reduced northward oceanic heat transport. Colder conditions extend into the lower atmosphere especially in winter and induce an anomalous blocking-like structure over northern Europe and associated anomalous easterlies over central Europe. In contrast to winter, the SPG slowdown has a weaker impact on the summer climate over Europe. Therefore, we have identified and examined high-resolution proxies and direct historical observations of climate and environmental conditions for consistency with these distinguishing traits of simulations with the SPG slowdown and with the timing of the Huaynaputina eruption.

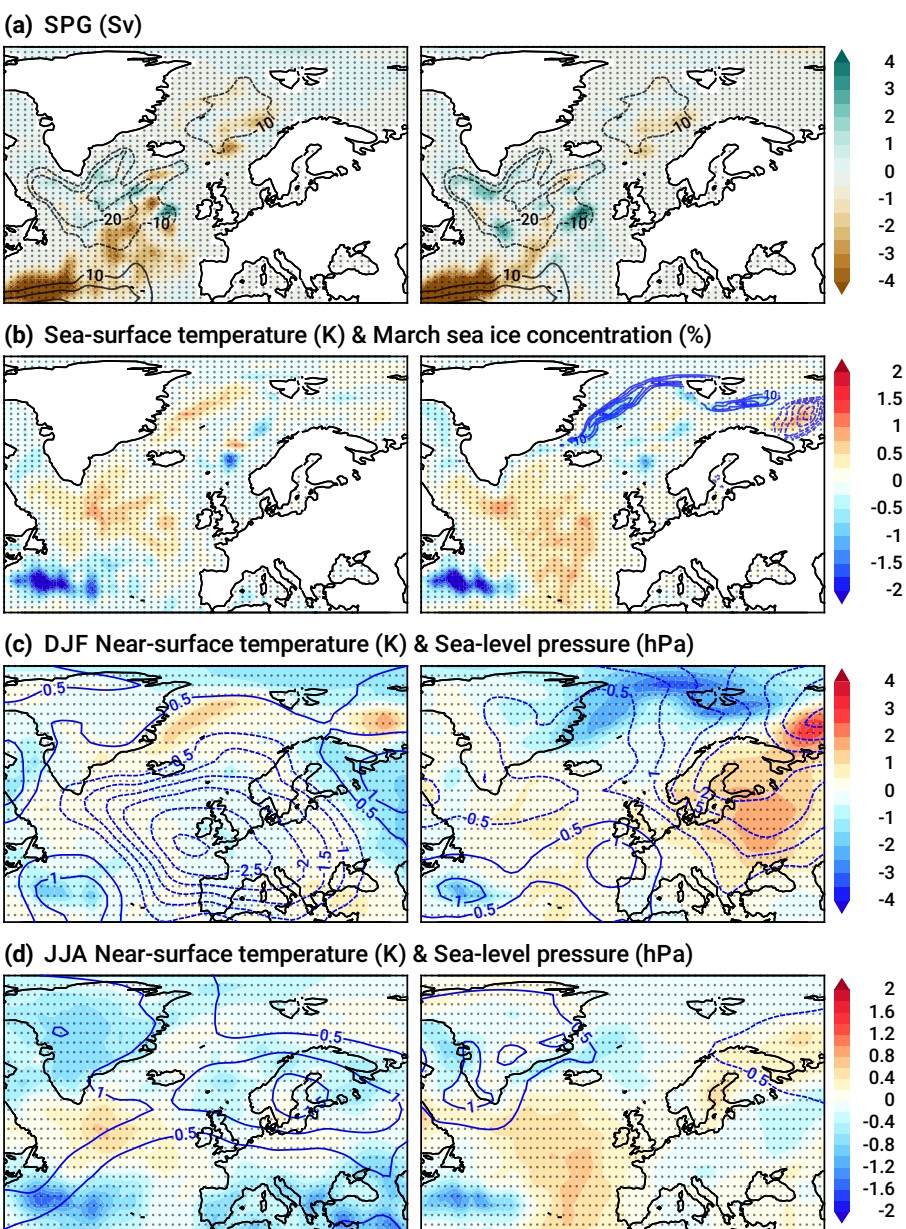

**Figure 3: Short-term response (anomalies in 1601-1602 w.r.t. 1550-1590) in the MPI-ESM-P ensemble with (left) and without (right) a subpolar gyre (SPG) shift. Anomalies are in (a) the annual barotropic streamfunction (in Sv; shading), with contours for the 1550-1590 mean; (b) annual sea-surface temperature (in K; shading) and March sea ice concentration (in percentage, contours every 5 %); and in near-surface (2 m) air surface temperature (in K; shading) and sea-level pressure (in hPa; contours), in (c) winter (DJF) and (d) summer (JJA). Stippling masks statistically non-significant anomalies at the 5% level based on a Student's t test.**

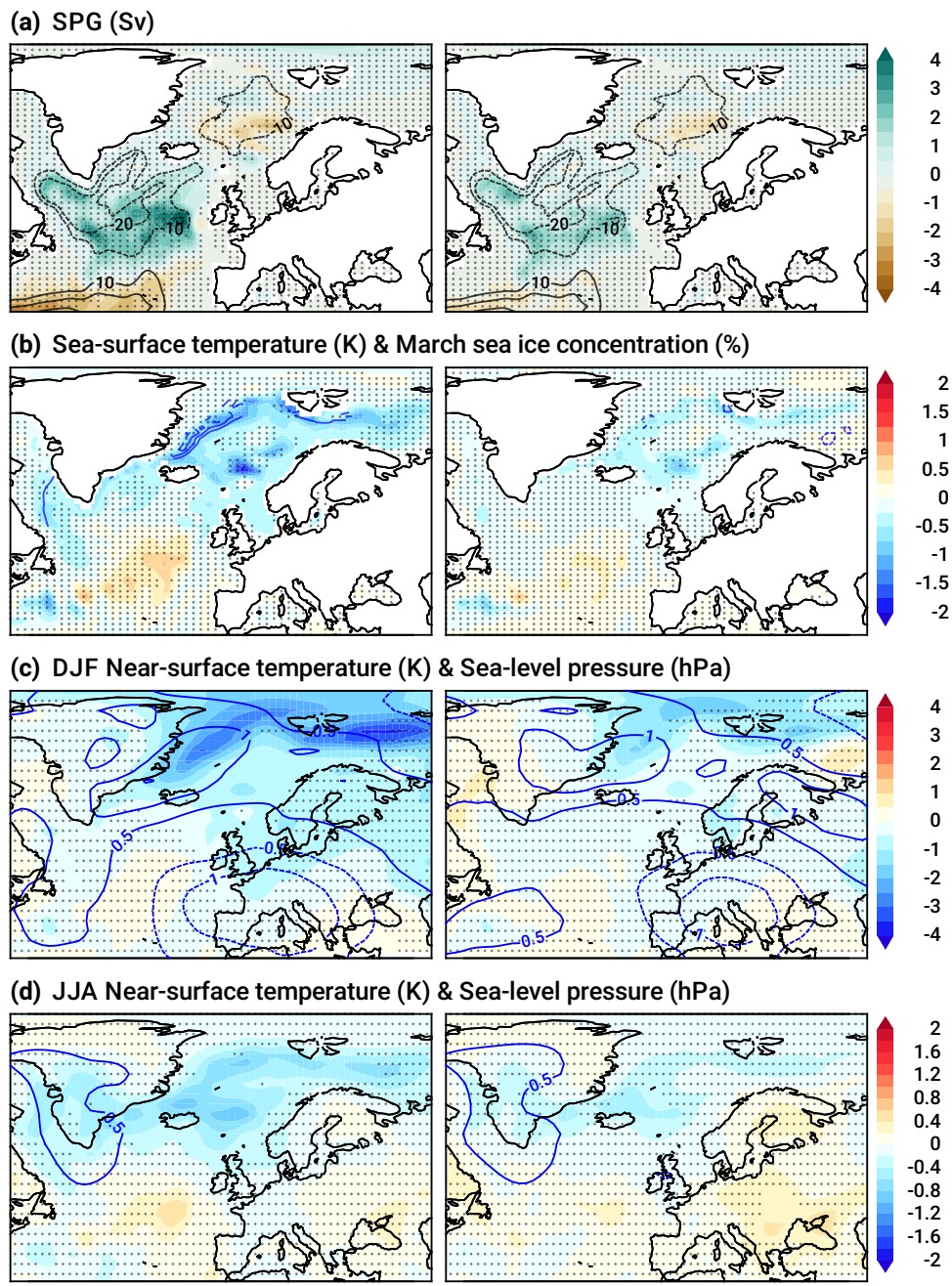

**Figure 4: Long-term response (anomalies in 1601-1630 w.r.t. 1550-1590) in the MPI-ESM-P ensemble with (left) and without (right) a subpolar gyre (SPG) shift, as in Figure 3.**

### 220   3. Annually Resolved Climatic and Environmental Reconstructions

This section presents the new high-resolution paleoclimate proxies and historical observations of climate and environmental conditions selected to determine whether these were consistent or not with a Huaynaputina eruption trigger for the previously identified SPG mechanism for persistent cooling. Past studies of the proposed mechanism relied mainly on reconstructions from marine archives with decadal to centennial resolution to determine whether historical changes in North Atlantic sea ice

extent and sea-surface temperature were compatible with an SPG shift (Moreno-Chamarro et al., 2017a, 2017b). Although these reconstructions did suggest that colder conditions and expanded sea ice prevailed over the first half of the 17th century in the North Atlantic, they lacked the temporal resolution to determine whether the SPG slowdown occurred through a regime

shift triggered by the 1600 Huaynaputina eruption. Similarly, past studies did not distinguish the different short-term and long-term impacts in the simulations as described above.

Therefore, in this study we first re-examine two high-resolution reconstructions of relevant climatic and environmental conditions: a network of tree-ring width and maximum density measurements as a proxy for summer temperatures, and the annual opening dates of ports around the Baltic as a proxy for winter sea ice and winter temperatures. We then present information from historical records containing direct observations of relevant conditions: wind directions from the North Sea coast and observations of sea-ice extent from Iceland and from ship voyages in the North Atlantic (see **Figure 5** for locations).

Although still sparse, such information provides additional constraints on the timing of climate and environmental changes in and around the North Atlantic, and hence it can be used to examine whether the post-Huaynaputina eruption period marked a shift in mean conditions for variables sensitive to the SPG slowdown.

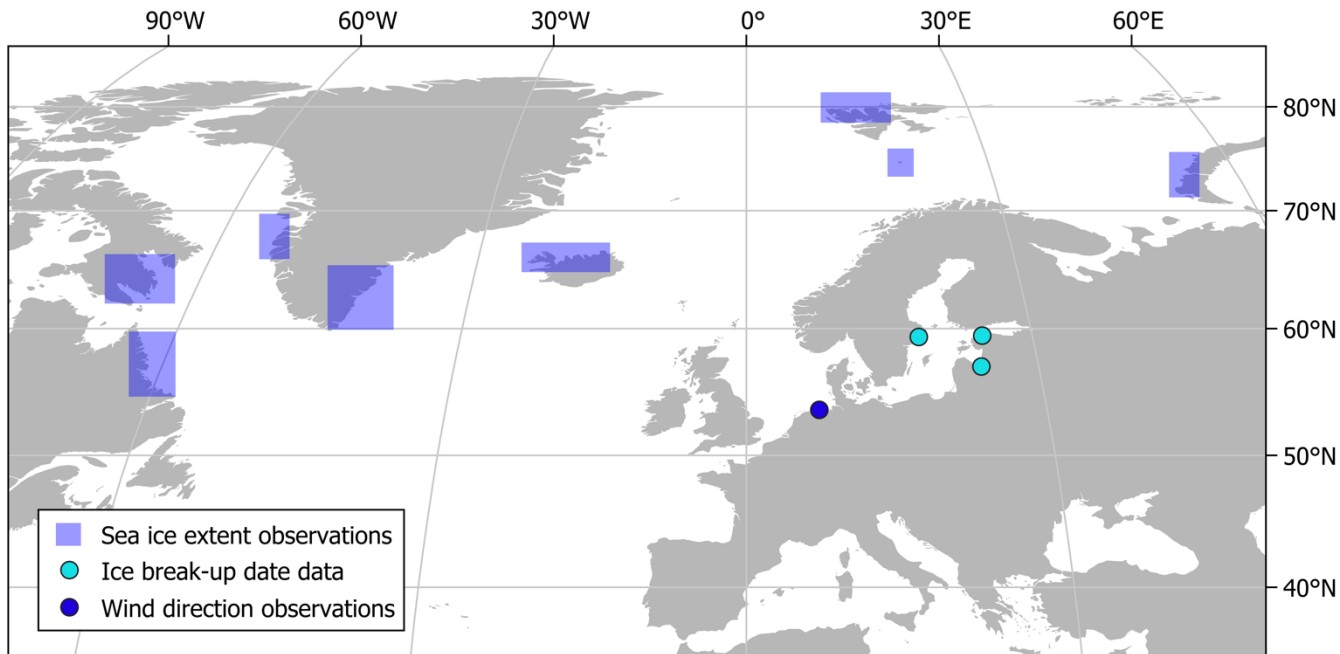

**Figure 5. Location of historical observations.**

**3.1 Gridded Summer Temperature from a Tree-Ring Width and Density Network**

The reconstruction of climatic conditions and cooling induced by the 1600 Huaynaputina eruption is based on the NVOLC v2 dataset (Guillet et al., 2017, 2020), which was created with the specific goal to detect and quantify volcanic cooling; it includes 13 tree-ring width and 12 maximum latewood density chronologies from across the NH (see Stoffel et al., 2015; Guillet et al., 2017, for details). The characterization of regional climatic change and the illustration of spatial temperature anomalies after

the 1600 Huaynaputina eruption was based on the NVOLC v2 network of tree-ring proxies and served the development of a climate field reconstruction of extratropical NH summer (JJA) temperatures as follows: In a first step, we grouped the 25 chronologies into 11 regional clusters using a correlation coefficient (r) threshold exceeding 0.3 over the period that was common to all chronologies (for details see Guillet et al., 2017, 2020). In total 3,486 grid points can thus be used for the reconstruction of the volcanic cooling induced by the Huaynaputina eruption. To examine short-term summer cooling induced

by the eruption, we reconstructed the temperature anomaly in 1601-1602 with respect to the average of the period 1550-1590. For comparison, we also employed the N-TREND (Northern Hemisphere Tree-Ring Network Development) spatial reconstruction of large-scale mean May-August temperature covering the Northern Hemisphere midlatitudes between 40° and 75°N (Wilson et al., 2016; Anchukaitis et al., 2017). The reconstruction is based on 54 published tree-ring records and uses different parameters as proxies for temperature, including ring-width (11 records), maximum latewood density (18 records), and mixed parameters (25 records) (see Wilson et al., 2016 for details). The N-TREND version used here is version (S) detailed

in Anchukaitis et al. (2017), which uses point-by-point multiple regression (Cook et al. 1994) of the tree-ring proxy records

available within 1,000 to 2,000 kilometers of the center point of each 5° x 5° instrumental grid cell and a similar nesting procedure to Wilson et al. (2016). We use the average of all the grid point reconstructions for the periods where the ice validation reduction of error is greater than zero.

### 3.2 Baltic Sea Ice Concentration and Winter Temperature

In the Baltic Sea area, port towns kept records when the harbors were free from ice, marking the beginning of the sea trade season following each winter (Jevrejeva, 2001; Tarand, 1992; Tarand and Nordli, 2001). In addition to direct observations, ice break-up dates can be obtained from indirect evidence such as harbor customs books, which marked the tolls paid on the date when the first ships arrived and departed after the winter (Leijonhuvfud et al., 2008; Degroot, 2018). These dates for ice break-up and the beginning of the sailing season have been identified as the proxy with the strongest reconstruction skill for Northern European winter and early spring temperatures (Hari et al., 2017); reconstructions based on this proxy can explain up to 67% of winter temperature variance (Leijonhufvud et al., 2010).

To assess short-term impacts of the 1600 Huaynaputina eruption, we compared the ice breakup data (Betin and Preobrazhinkski, 1959; Leijonhufvud et al., 2010; Tarand et al., 2013) and simulations for the years 1599 and 1601, each with respect to a 1550-1590 reference period. In the model, breakup dates were defined as the first day of the year when sea-ice concentrations in the Baltic Sea (averaged between 10°E-30°E and 50°N-60°N) were lower than 1%, after smoothing with a 7-day running mean. This threshold ensures similar dates in the model and historical records, around early April. To assess whether there was a long-term shift following the Huaynaputina eruption, we looked for changepoints in the Tallinn harbor ice breakup timing data for any year within any segment length of ≥30 years in the period 1550-1675 using the BinSeg and PELT methods for changepoint detection (Scott and Knott, 1974; Killick et al., 2012; Killick and Eckley, 2014). The changepoint analysis could not be performed with the data from Stockholm and Riga harbors due to considerable data gaps before and after 1601.

### 3.3 North Sea Wind Direction

Anemometer readings are unavailable for the late 16[th] and early 17[th] centuries. While numerous voyages recorded information in ship logs, these do not provide a continuous series for wind direction over the North Sea until later in the 17[th] century. However, this period saw the beginnings of daily weather narratives in journals and almanacs, including some records with consistent and reliable observations of wind direction (Pfister et al., 1999; Pfister and White, 2018). Useful for the target region are the weather journals of David Fabricius, compiled in East Friesland (Germany) during 1590-1612 and reproduced and analyzed in a previous study (Lenke, 1968). These observations were recorded in flat terrain and therefore observations should reflect regional wind direction rather than local obstructions. We compared the percentage of days in each year with winds from each direction in the reconstruction and in the MPI-ESM-P ensemble of simulations with and without the SPG shift.

### 3.4 North Atlantic Sea Ice Extent

Previous studies have utilized proxies in natural archives to reconstruct North Atlantic sea ice extent at multi-decadal to decadal resolution. In particular, studies of IP$_{25}$, a lipid biomarker produced by diatoms that inhabit the ice, indicate a significant expansion of spring sea-ice extent ca. 1600 on the Icelandic shelf (Massé et al., 2008; Cabedo-Sanz et al., 2016), the east Greenland shelf (Kolling et al., 2017), and in the Fram Strait (Müller et al., 2012). Historical observations, which are precisely dated, may indicate whether the shift occurred after the 1600 Huaynaputina eruption in a manner consistent with an eruption-triggered SPG shift.

This study examines three sources of sea-ice observations for the period. First, consistent observations of sea ice around the north shore of Iceland began during the late 16[th] century. These have been employed in several past studies as an indicator of

annual sea-ice extent as well as local temperatures and societal impacts in Iceland (Ogilvie, 1995; Ogilvie and Jónsdóttir, 2000; Ogilvie and Jónsson, 2001; Ogilvie, 2019; Ogilvie, in press). Second, during the first two decades of the 17th century, walrus-hunting expeditions to Bjørnøya, sponsored by the English Muscovy Company, spurred the development of a large European whaling industry that centered on the bays of Spitsbergen. Because sailors in search of walruses and whales initially hunted from temporary encampments along the coast, expeditions usually arrived at their hunting grounds as soon as sea ice began to retreat from these bays (Degroot, 2020).

Third, an increasing number of European ships sailed into the North Atlantic and Arctic seas on voyages of exploration. Although this period predates systematic reconstruction of sea-ice extent from ships' logs, many of these voyages left detailed records, including descriptions of sea ice. Because written observations from the 16th and early 17th centuries lacked precise measurements or a standardized vocabulary for ice conditions, the most objective criterion for determining sea-ice extent is when and where ships changed course due to sea ice described as "impassable" or a similar term. For consistency, we identified common points at which ships did or did not turn back from an intended destination due to dangerous sailing conditions, using only those locations that had records in the decades before and after 1600. Decisions to turn back a voyage could depend on a mix of factors, including cold, duration of voyage, type of ship, crew morale, and danger of sailing conditions. Therefore, we analyzed voyage journals, ship logs, and other underlying information to identify important factors in each case (see Supplement).

## 4. Results

### 4.1 Northern Hemisphere Summer Temperature

As found in previous tree ring-based reconstructions, the 1600 Huaynaputina eruption produced a sharp decline in NH summer temperatures during 1601-1602 (**Figure 6**). In fact, with a cooling of -1.6˚C in 1601 CE, the Huaynaputina eruption caused the most significant cooling recorded in the Northern Hemisphere reconstruction over the past 1500 years. The simulations and NVOLC v2 reconstruction show agreement in the magnitude of the average cooling over Europe; however, there is little agreement in the spatial pattern of the anomaly, even in the closest simulation (**Figure 7**). The reconstruction-simulation discrepancy is especially evident over Scandinavia and the Baltic region, where simulations yield much more cooling compared to the reconstruction.

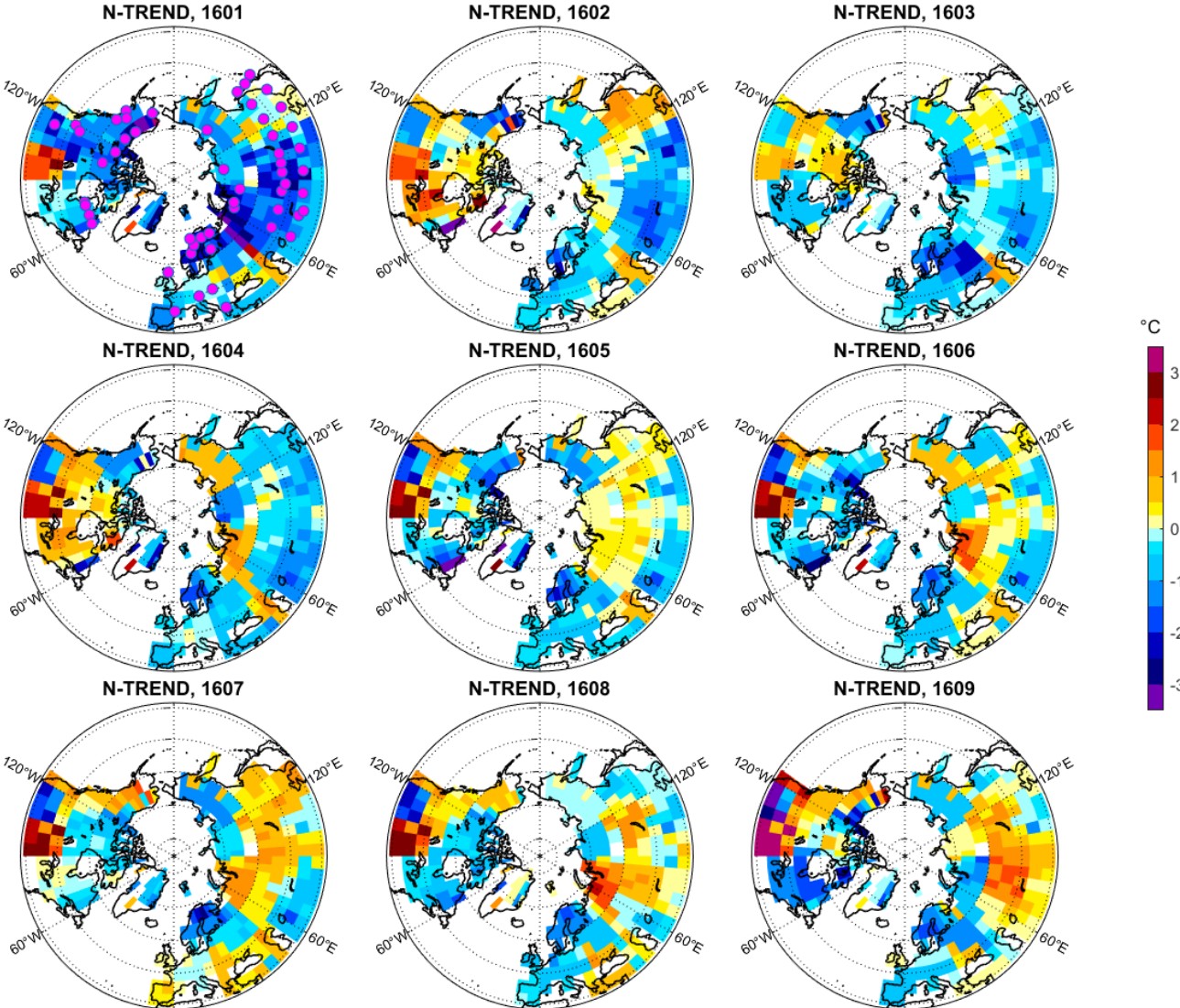

**Figure 6. N-TREND temperature anomaly in each year of the period 1601-1609 with respect to the average over the period 1550-1590. The purple dots in top-left plot indicate sites of tree-ring width and maximum latewood density used for the spatial reconstruction. The size of reconstructed grid cells is 5°x5° lat/long. The instrumental data used as target field for the reconstruction are May to August monthly temperature anomalies w.r.t. the 1961-1990 period extracted from the HadCRUruT4 (Cowtan and Way, 2014). The spatial reconstruction was developed using a point-by-point regression (Cook et al., 1999) which accounts for the spatial distribution and relationship of the proxy predictor network to the target field.**

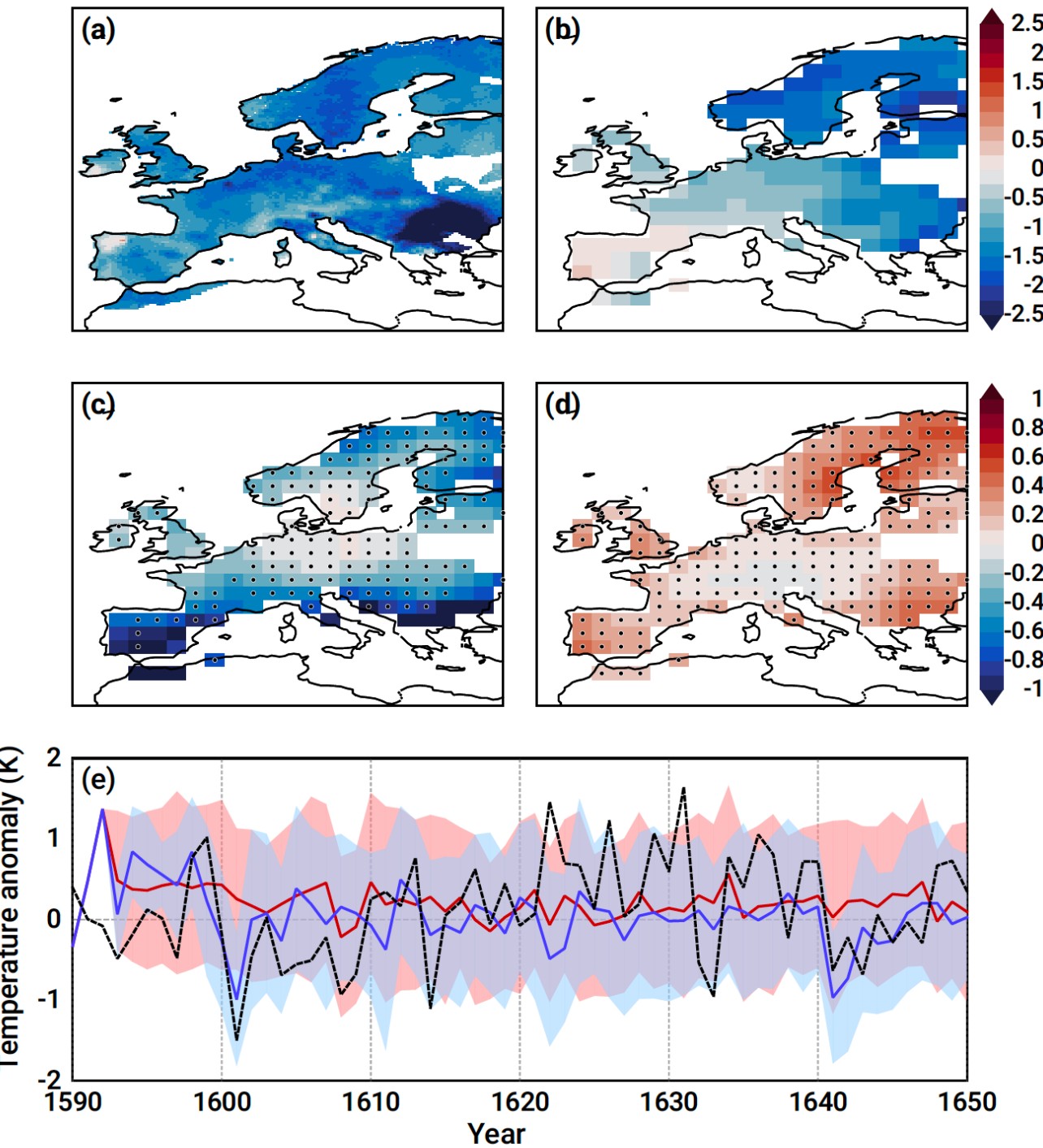


**Figure 7: (a-d) Spatial pattern of the 1601-1602 anomaly in summer (JJA) temperature (in K; w.r.t. 1550-1590) in (a) the NVOLC v2 reconstruction; (b) the MPI-ESM-P ensemble member r3i8, for which the spatial correlation coefficient with the reconstruction in (a) is the largest (0.17) with the reconstruction in (a); (c) the ensemble with volcanic forcing; and (d) the ensemble without volcanic forcing between 1593-1650. The r3i8 shown in (b) is a sensitivity simulation branched in 1600 from the original simulation r3i1p1**

**through a slight, temporary perturbation in one atmosphere model's parameter (Moreno-Chamarro et al., 2017b). Grid points without stippling in (c) and (d) are where at least 80% of the ensemble members agree on the sign of the anomaly. (e) Summer temperature anomaly (in K; w.r.t. 1550-1590) of the NVOLC v2 reconstruction (black, dashed line; shifted +0.5 K) and of the ensemble with (blue) and without (red) volcanic forcing (for which thick lines are the ensemble mean and the shading is the ensemble standard deviation).**

**4.2 Baltic Sea-Ice Concentration and Winter Temperature**

As shown in **Figure 8**, the ice break-up data in Riga, Stockholm, and Tallinn suggest longer "winter seasons" in 1599 and 1601 compared to the period 1550–1590, about 1-2 weeks longer in 1601 in Riga and Stockholm than in Tallinn. This may reflect the volcanic cooling over Europe after 1600. In the sensitivity simulations, however, ice break-up in the Baltic Sea in 1601 can occur at a date both earlier and later than in 1599 with and without the 1600 Huaynaputina eruption. The simulations

therefore do not support a robust connection between the volcanic cooling and a later ice break-up date in these cities immediately after the eruption. This is consistent with the results described above on the simulated short-term climatic response (Figure 2), where the volcanic cooling is (on average) strongest in summer and mostly absent in winter over the region. This result also confirms that intrinsic climate variability, including climate modes such as the NAO and Arctic Oscillation, plays a major role in setting winter conditions over Europe, including Baltic Sea temperatures and sea ice (Moreno-

Chamarro et al., 2017b; see also Chen and Hellström, 1999; Omstedt and Chen, 2001; Eriksson et al., 2007; Zanchettin et al., 2019). In particular, the duration of Baltic sea ice and port closures shows a negative correlation with the NAO index (Jevrejeva, 2002); thus, the absence of a post-eruption positive NAO anomaly helps account for the longer port closure dates ca. 1600.

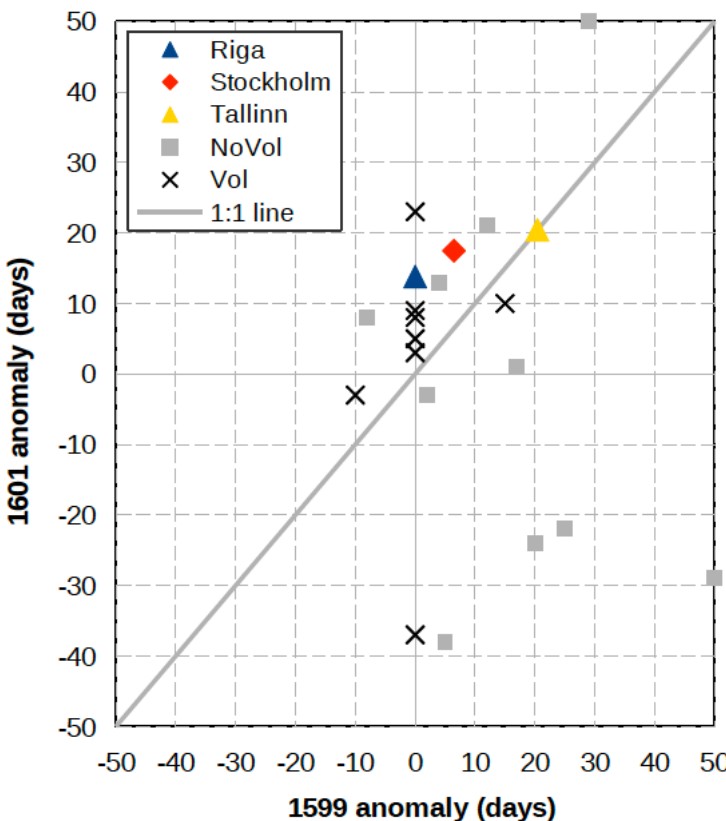

**Figure 8. Anomaly in the sea ice breakup date in 1601 and 1599, w.r.t. the 1550-1590 period. In the MPI-ESM-P simulations, breakup dates are defined as the first day of the year when sea-ice concentration in the Baltic Sea (averaged between 10˚E-30˚E and 50˚N-60˚N) is lower than 1%, after smoothing with a 7-day running mean. This threshold ensures similar dates in the model and historical records, around early April. A breakup anomaly equal to 50 indicates that the simulation does not show a breakup date in a year. Note that 6 simulations have the same 1599 anomaly, since the ensemble begins in 1600.**


As shown in **Figure 9**, changepoints in Tallinn harbor ice break-up dates were detected in 1597 and 1630. The change in mean of the Tallinn ice break-up data time series indicates colder winters and greater persistence of Baltic sea ice in the period 1597-1630 than during 1563-1596 or 1631-1664. Similarly, the winter ice severity index from the southwestern Baltic identifies a phase of increased severity between 1593 and 1630 (Koslowski and Glaser, 1999). The timing raises three at least possibilities:

First, the winter cooling may have been unrelated to the SPG slowdown. Second, the cooling may have been caused by an SPG slowdown that began before the Huaynaputina eruption. Finally, there may have been other causes for cooling during

the 1590s—such as the 1591 and 1595 eruptions described above—before the Huaynaputina eruption triggered an SPG slowdown and further cooling. Thus, the timing of winter cooling detected in this reconstruction could be consistent with an eruption-induced SPG slowdown but does not provide further evidence for it.

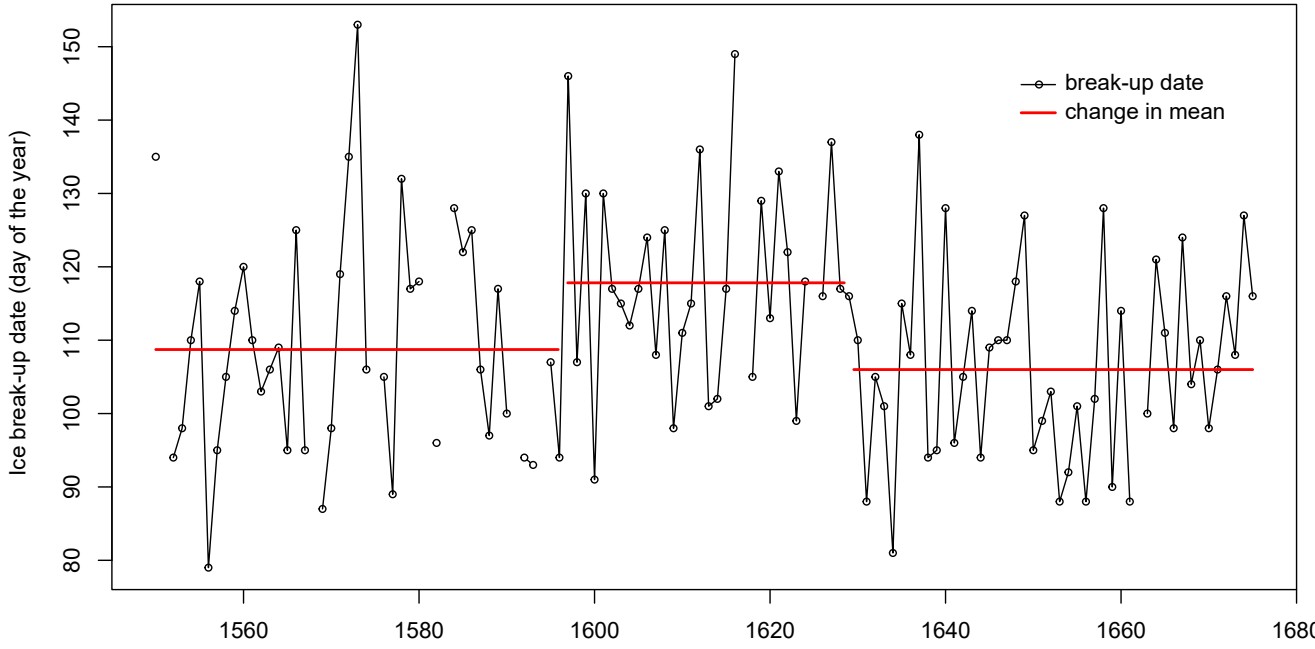

**Figure 9. Tallinn ice break-up dates 1550–1670 and changepoints in mean.**

### 4.3 North Sea Wind Direction

Observations in contemporary weather diaries indicate a marked reduction in the frequency of southwesterly winds and an increase in the frequency of easterly winds in the period 1590-1612 over the North Sea as compared to an early instrumental reference period, 1881-1925 (Lenke, 1968). The greatest change occurred during winter months. During the modern reference period the highest frequency of easterlies near East Friesland occurred in May-June, when they were predominant on only 10-15% of days. Averaged over the years during which David Fabricius kept daily observations, easterly winds were predominant on more than 20% of days during October, December, February, and March. As noted by Metzger and Tabeaud (2017), Fabricius's observations also indicate far more frequent snows and longer frosts during most of the winters in this period than those observed since the 20th century.

The year 1601 has the highest frequency of days with winds predominantly out of the north, northeast, or east. As displayed in **Figure 10**, this short-term shift toward more frequent northerlies is found in simulations with the SPG shift but not in those without it. Otherwise, neither the reconstruction data nor simulations provide a clear signal of volcanic forcing or the SPG shift in the characteristics of winter winds over the North Sea.


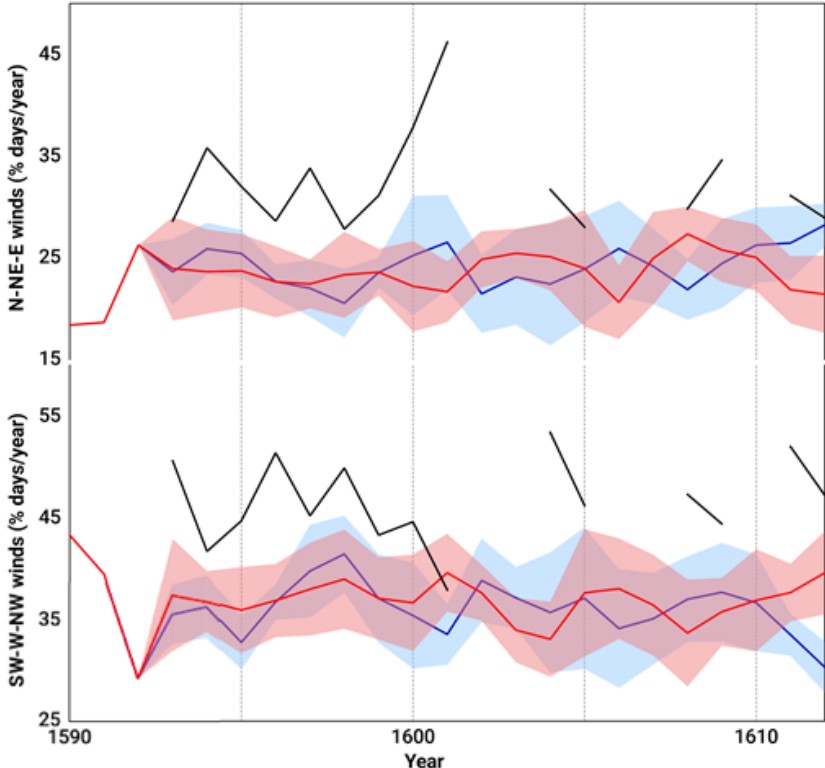

**Figure 10. Percent of days in a year with winds predominantly from the North, Northeast, and East (top) and from the Southwest, West, and Northwest (bottom) in the historical data (black) and in the MPI-ESM-P ensemble of simulations with and without a subpolar gyre (SPG) shift (blue and red respectively; with the ensemble mean as the solid line, and standard deviation in shading).**

**4.4 North Atlantic Sea Ice**

**Table 1** summarizes all observations that met the criteria listed in section 3.4. Where multiple sources were available for a single location and year, they agreed in all cases, with the exception of mixed reports in 1612 at Bjørnøya and 1614 and 1617 around northern Spitsbergen, probably due to shifting ice conditions within the same season. The small number and heterogeneity of the observations do not permit a robust reconstruction of sea-ice conditions; however, the results support the proxy evidence in natural archives indicating an increase in North Atlantic sea ice after ca.1600. During the 1570s-1590s

several voyages were able to approach the southern tip of Greenland from the east or reach western Novaya Zemlya. Two of three could land at Labrador and four of five at Baffin Island. The Icelandic historical sources suggests that the 1560s were very cold with much sea ice but that the 1570s were mild; they provide indications that there was much sea ice off the coasts during the 1580s and years of the 1590s. From the period 1602-20, no voyages were able to land on the southern tip of Greenland, the northern Labrador coast, or Novaya Zemlya, and only two out of five reached Baffin Island. Icelandic observers

did not report any years free of sea ice around Iceland and recorded several years with extensive sea ice. A few years, including 1602 and 1615, were notable for descriptions of severe sea ice in both ship observations and Icelandic records. Conditions around Spitsbergen and Bjørnøya remained variable.

Thus, results from historical observations support a change in sea-ice conditions during the early 17[th] century consistent with simulations of a post-eruption SPG slowdown. However, the precise timing of the sea-ice shift and its association with a

change in SPG strength remain ambiguous. On the one hand, it could be the case that post-eruption cooling generated a short-term expansion in sea ice after 1600, including the severe conditions observed in 1602, and only then did an eruption-triggered SPG slowdown initiate a long-term change in sea-ice conditions. On the other hand, the apparent increase in sea ice in 1602-20 with respect to 1570-1590s would also be consistent with a pre-eruption onset of the SPG slowdown.

| Voyages of Exploration | Pre-eruption (1570-1600) | | Post-eruption (1601-1620) | |
|---|---|---|---|---|
| | Dates passable | Dates impassable | Dates passable | Dates impassable |
| **SE Greenland** | 6/1578, 7/1587 | 7/1576, su1577, 7/1585, | -- | 6-7/1602, 6-7/1605, |

| | Light/no sea ice | Heavy/late sea ice | Light/no sea ice | Heavy/late sea ice |
|---|---|---|---|---|
| | | 6-7/1586 | | 6-7/1606, 6/1607, 6/1610, 5/1615 |
| **W Greenland** | 8/1585, 7/1586, 7/1587 | 9/1576 | 7/1605, 8/1606, 7/1616 | -- |
| **Baffin** | 8/1576, su1577, 8/1585, 8-9/1586 | 7/1578 | 7/1610, 7/1611 | 7-8/1602, 7/1606, 8/1616 |
| **Labrador** | su1577, 8/1586 | 7/1576 | -- | 8/1602, 7/1606 |
| **W Novaya Zemlya** | 8/1596, 8/1597 | 7-8/1594 | -- | 6-7/1608 |
| **Bjørnøya** | -- | 6/1596 | -- | 5/1612 |
| **Spitsbergen** | -- | 6-7/1596 | su1612 | 7/1607, su1614 |

| **Sea Ice Descriptions** | Pre-eruption (1570-1600) | | Post-eruption (1601-1620) | |
|---|---|---|---|---|
| | **Light/no sea ice** | **Heavy/late sea ice** | **Light/no sea ice** | **Heavy/late sea ice** |
| **Iceland** | 1570, 1572, 1592, *early 1590s* | *1580s* | -- | 1602, 1604, 1605, 1608, 1612, 1615, 1618 |
| **Bjørnøya** | (no data) | | 1603, 1608 | 1604, 1605, 1606, 1609 |
| **Spitsbergen** | (no data) | | 1612, 1617, 1619 | 1611, 1614, 1615, 1616, 1618 |


**Table 1. Summary of sea-ice observations from written sources. Numbers before years indicate months of year; sp[ring] = AMJ; su[mmer] = JAS. All dates converted to Gregorian calendar. Entries in italics indicate indirect observations or implied conditions based on information in chronicles. See Supplement for underlying sources and information**

**5. Discussion**

**5.1 The Huaynaputina Eruption as a Possible Trigger of 17th-Century Persistent Winter Cooling**

Our results, although consistent with the hypothesis that the Huaynaputina eruption triggered an SPG shift, do not provide additional support for this hypothesis. As discussed in section 2, simulations with an SPG shift showed characteristic short-term summer cooling and long-term winter anomalies, including persistent cooling and sea-ice expansion. The large degree of short-term summer cooling in the NVOLC v2 reconstruction is similar to the mean of simulations with an SPG shift.
Nevertheless, the spatial pattern of the anomaly differs between the reconstruction and those simulations. There are several possible sources for this discrepancy. The NVOLC v2 reconstruction has weaker spatial coverage in southeastern Europe, where some of the strongest differences appear (Guillet et al., 2017). The climate model necessarily simplifies climatic processes. Most importantly, this study has considered a relatively small ensemble size (10 realizations with volcanic forcing and 10 without) within a single model, as well as a single reconstructed history of volcanic forcing. Beyond limitations due to
the specificity of the chosen model and forcing, the ensemble size seems to be insufficient to encompass the range of possible climate responses to the Huaynaputina eruption that stems from their dependency on the initial state of the climate system at the time of the eruption.

Regarding winter anomalies, historical observations of North Sea wind direction indicated an unusually high frequency of winter northerly to easterly winds throughout the decade before and after 1600, especially in 1601. However, the series was
too short and incomplete to demonstrate a long-term shift following the eruption. The Baltic harbor dates also indicated a short-term anomaly in 1601 as well as a multi-decadal period of colder, icier conditions. However, the change-point analysis indicates that this transition to colder icier conditions began before the Huaynaputina eruption, rather than several years following the eruption, as would have been expected with an eruption-triggered SPG shift. The historical observations of North Atlantic sea ice, taken in conjunction with previous paleoclimate studies discussed in section 3.4, appear to confirm this timing.
In general, observers found icier conditions starting ca.1600, rather than another decade following the Huaynaputina eruption. Therefore, the reconstructions are consistent with at least two climatic scenarios, each found in different sets of simulations. In the first scenario, the 1600 Huaynaputina eruption triggered the SPG slowdown, but the shift to colder and icier conditions in Northern Europe and the North Atlantic had already begun by 1600 due to intrinsic climate variability or a different externally forced mechanism. This latter possibility stems from the fact that the 1600 Huaynaputina eruption was only the final
eruption of a volcanic cluster that started in 1585. In the second scenario, the SPG slowdown commenced by 1600 without

any role for the Huaynaputina eruption. Furthermore, it is possible that none of the simulations has reproduced the mechanism for persistent cooling that operated in the real world. Thus, our examination of high-resolution proxies and observations neither disproves nor confirms an eruption trigger for the previously proposed SPG-shift mechanism.

Our results, although inconclusive, suggest two ways forward. First, further comparison between high-resolution reconstructions and a larger ensemble of climate simulations could improve the chances for determining whether or not the 1600 Huaynaputina eruption triggered the SPG slowdown. With a larger ensemble, it may become possible to identify an initial seed for the SPG slowdown mechanism, since the signal-to-noise ratio of emergent features increases with the ensemble size. Identification of such an initial seed in proxy reconstructions and historical observations may enable more certain identification of an SPG slowdown and its causes than the attempts to find characteristic effects of such a slowdown in this study. This type of study may require large ensembles of simulations (20 members or more) in order to detect a clear signal for the onset and evolution of a climate mechanism above the noise of interannual climate variability. If a larger ensemble were insufficient to determine an initial seed, then a higher-resolution model might also prove necessary, or else a model that could better represent the volcanic forcing (e.g., with updated aerosol parametrizations or updated volcanic forcing histories) and other external forcings (such as solar). Furthermore, with higher-resolution reconstructions, future studies might specify initial climate conditions in the model world closer to those of the real world.

Second, additional high-resolution climate proxies and historical records covering more locations in the North Atlantic could help determine whether anomalies during the late 1590s were indicative of an SPG slowdown preceding the 1600 Huaynaputina eruption. Moreover, forward modeling might be used to directly simulate additional proxies and conditions present to contemporary observers in order to provide stronger tests of an eruption trigger and SPG shift using historical climatology data. However, such modeling may have to take into account not only physical processes but human processes of observation, recording, and transmission.

The results also reveal strengths and weaknesses in the use of historical records to assess model-derived mechanisms of climate variability. Compared to proxies in natural archives, information drawn from archives of societies often has greater specificity and resolution but less continuity and homogeneity (Brönnimann et al., 2018). These sources may thus be more suitable for testing the presence of specific initial conditions than identifying spatial patterns in anomalies or long-term climatic trends, with the possible exception of very consistent and precise historical sources such as the Baltic harbor dates. Historical climatology may also contribute more to the analysis of climatic events and changes during the 18[th] and 19[th] centuries, for which there are more consistent and widespread historical observations, than those found in earlier centuries.

**5.2 Implications for the History of Climate and Society**

As described in the introduction, previous studies have identified widespread mortality crises and conflict in Europe during the 1590s and early 17[th] century associated with cool, wet summers and resulting harvest failures. Our results place the occurrence of these exceptionally cool summers in Europe and around the North Atlantic following the 1595 Nevado del Ruiz 1600 Huaynaputina eruptions into a wider context of climatic change. In addition, our results indicate the onset of colder winters and more sea ice in the North Atlantic by 1600 and lasting into the 1620s, thus preceding the eruption and persisting through and beyond the post-eruption summer cooling.

These long, cold and snowy winters can affect human societies in many ways, especially in more marginal areas of agriculture. In areas where cattle were kept on winter pastures, such as parts of Ireland, severe winters could kill stock and reduce births (Ludlow and Crampsie, 2018). In addition, in high-latitude and altitude agricultural areas where the animals had to be kept indoors over winter, such as Northern Europe and the Alps, farmers may have run out of fodder during prolonged winter seasons, resulting in reduced dairy production or emergency sale or slaughter of animals (Soininen, 1974; Pfister, 2005). Long snowy winters also posed a risk to winter grains, since heavy long-lasting snow cover created optimal conditions for snow mold fungi (*Microdocium nivale*) to damage seedlings wintering under the snow cover (Pfister, 2005; Solantie, 2012).

Furthermore, in northernmost Europe, increased snow depth and prolonged winter delayed the start of the growing season and thus postponed the harvest to less favorable times in late August or early September, when the first autumn frosts commonly occurred (Huhtamaa et al., 2015). In Finland, contemporaries witnessed this delay in 1601. That year, in southern parts of the country, the amount of harvested grain barely exceeded the amount sown, and further north the harvest was lost altogether due to the autumn frost (Voipio, 1914; Huhtamaa et al., 2020).

The agricultural hardships of the persistent winter cooling culminated in widespread famine in 1601 in the Swedish Realm (roughly the areas of present-day Sweden, Finland and Estonia) (Lilja, 2006; Seppel, 2014; Huhtamaa, 2018). However, the cold alone does not solely explain the human suffering, since the whole northern Baltic region experienced political instability and distress at this time. King Sigismund and Duke Charles fought over the throne in the late 16[th] century, western Finland underwent a peasant uprising (1596-1597), and Estonia was the battleground of warfare between Sweden and Poland from 1600-1611 (Seppel, 2014; Huhtamaa, 2018). These existing conditions increased social vulnerability across the Swedish Realm and arguably exacerbated the human consequences.

Historical case studies demonstrate that these multidecadal changes, characteristic of the SPG slowdown mechanism described in this study, also featured in societal impacts and adaptations in the Arctic. Extensive but fluctuating sea ice redirected expeditions led by captains such as Willem Barents and Henry Hudson towards lucrative sites for colonial exploitation, such as the bowhead whale feeding grounds off Svalbard and the Hudson River (Degroot, 2015a; Degroot, 2015b; White 2017). When European whalers competed for access to preferred whaling locations along the coast of Svalbard and the relatively nearby island of Jan Mayen, extensive sea ice discouraged conflict by either separating whalers from one another or by concentrating bowhead pods in just a few of Svalbard's many bays (Degroot, 2020). In European waters, winter sea ice suffocated seaborne trade but provided new possibilities for transportation and encouraged new transportation technologies in the densely populated coastal regions of the Dutch Republic (Degroot, 2018).

Recurring cold winters and sea-ice expansion had significant repercussions for early European colonization in eastern North America as well. Expeditions often arrived poorly supplied and vulnerable to extreme weather. Early and late frosts limited planting and harvesting of crops; long freezes and snow cover limited foraging; and early winters and late springs made for long periods without fresh food, contributing to deadly outbreaks of scurvy. Attempted settlements at Tadoussac, Quebec (1601); St. Croix, Maine (1605); and Sagadahoc, Maine (1608) were each abandoned in less than a year after experiencing extremely cold winters. The first settlers at Jamestown, Virginia (1607) and Quebec (1608) barely survived hunger and scurvy, respectively. The poor reputation of northern colonies as well as expansion of sea ice and difficult sailing conditions may have diverted interest and investments into more southern colonies during the 17[th] century (White, 2017; Zilberstein, 2016). During the same period some indigenous nations of today's northeastern US and eastern Canada migrated south while others adapted their hunting and horticulture to longer and colder winters (Hall, 2015; Wickman, 2018).

## 6. Conclusions

This study has examined high-resolution proxies and historical observations to investigate whether the 1600 Huaynaputina eruption triggered persistent cooling in the North Atlantic region by initiating an SPG slowdown mechanism identified in previous modeling studies. Although the high-resolution reconstructions and historical observations of summer and winter temperature, wind direction, and sea-ice extent are consistent with such an eruption-induced mechanism, the results are inconclusive, particularly since reconstructions and observations indicate that the onset of winter cooling and increased sea ice may have preceded the Huaynaputina eruption.

By identifying potential strengths and weaknesses in the use of historical climatology for testing model-derived climate mechanisms, our assessment may guide future research both for the specific case of the early 17[th]-century climate shift and for other episodes of paleoclimate variability. Our study underlines the potential of historical climatology to reconstruct highly resolved local climatic and environmental conditions relevant to studies of model-derived mechanisms. Moreover, the

reconstructions and observations presented in this study have helped clarify human dimensions of climate variability during this period, including roles of short-term post-eruption anomalies as well as longer-term cooling and sea-ice expansion in societal impacts and adaptations.

## Author Contributions

SW, EM, and DZ designed the study. EM and DZ provided paleoclimate simulations and analysis. MS, CC, HH, SW, and DG provided paleoclimate data and historical observations. SW, DG, and HH provided societal impacts and adaptation analysis. All authors provided figures, discussed methods and results, and commented on the manuscript.

## Competing Interests

The authors declare that they have no conflict of interest.

## Acknowledgements

This study is a product of the PAGES-VICS and PAGES-CRIAS working groups. We thank Astrid Ogilvie for providing information on Icelandic sea-ice observations and commenting on the manuscript. HH's work was supported by the Swiss National Science Foundation (grant no. P2BEP1_175214). MS and CC received funding from the Swiss National Science Foundation through the SNSF Sinergia CALDERA project (grant agreement no. CRSII5_183571). EM acknowledges funding from the Spanish Science and Innovation Ministry (Ministerio de Ciencia e Innovación) through the STREAM project (PID2020-114746GB-I00) and from Fonds de la Recherche Scientifique – FNRS and the FWO under the Excellence of Science (EOS) program through the PARAMOUR project (grant no. O0100718F, EOS ID no. 30454083).

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
