# Peer review of "The 1600 Huaynaputina Eruption as Possible Trigger for Persistent Cooling in the North Atlantic Region"

_Climate of the Past, 2021_

## Author Comment (AC1)

**RC1**

The lead author is one of the world's key scholars in Historical Climatology and Climate History cooperating with natural scientists and historians, mainly from Europe. The Huaynaputina is a stratovolcano in southern Peru. Its eruption in February 1600 CE triggered a persistent summer and winter cooling in the North Atlantic region during the early 17. It-was the largest eruption in the Andes in historical times. The paper explores the eruption-induced cooling mechanism in some detail. The study compares simulations and a North Atlantic subpolar gyre shift in annual proxies in archives of nature to more detailed with documentary evidence.

The Huyaputina injected fewer sulfates into the stratosphere than other LIA eruptions with a compatible cooling effect. In particular the eruption generated much more and longer lasting winter cooling than summer cooling in Central and Northern Europe, which is unusual. It is hypothesized that reduced heat transport by the North Atlantic Subpolar Gyre SPG in terms of a cooling of the North Atlantic may be one of the reasons, though many aspects remain unclear. Historical written records as well as contemporary historical observations of relevant climate and environmental conditions demonstrate patterns of cooling and sea ice expansion consistent with, but not necessarily indicative of an eruption trigger for the proposed SPG slowdown mechanism

The arguments of scientists and modellers should still be improved in view of the limited understanding of people from the historical sciences for processes in the ocean. Some relevant studies might still be included: The herring catch on the west coast of Scotland declined remarkably between 1585 and 1597 which Parry (1978) interpreted this as an escape of this cold-sensitive species from the cold water masses advancing southwards. The detailed temperature reconstruction by Dobrovolny et al. (2010) for Western and Central Europe since 1500 CE was overlooked. It contains seasonal temperature that are explained in more detail by the recent synthesis by Pfister and Wanner (2021) that is published on 6th September. These data show that the very severe winter of 1600 preceded the Huynaputina eruption in contrast to the cold winter in 1601. Likewise, just spring 1600 was very cold. On the other hand, the summer 1601 was very cold.

We thank the author for his generous comments on the manuscript. We will add discussion of previous historical climatology studies relevant to the timing of winter cooling and changing North Atlantic currents to the paragraph in lines 73-79. The fish catch data in Parry 1978—as well as Jürgen Alheit and Eberhard Hagen, "Long-Term Climate Forcing of European Herring and Sardine Populations," *Fisheries Oceanography* 6 (1997): 130–39—indicate declining herring catch at the north end of their range and expansion of the south end of their range by the 1590s, which would be consistent with our finding of cooling preceding the 1600 eruption; although as discussed in Poul Holm et al. "The North Atlantic Fish Revolution (ca. AD 1500)." *Quaternary Research*, 2019, larger shifts in the fishing industry, including the abundance of previously unexploited fisheries being discovered off North America, make it difficult to attribute fluctuations in European fish catch during the late 1500s directly to climatic changes. We will also include these indications of changing ocean currents from historical fish catch in the paragraph currently in lines 73-79. The Moreno-Chamarro et al. 2017 studies presenting the SPG shift mechanism already considered the Dobrovolny et al. 2010 reconstructions (as well the Luterbacher et al. 2004 reconstruction). We will include these studies, too, when discussing the onset of colder winter temperatures over Europe during the 1590s.

---

## Author Comment (AC2)

**RC2 with responses**

The manuscript fits the scope of the journal well with integration of climate modeling, paleoclimatology, and historical climatology to address the effect of the Huaynaputina (Peru) eruption in 1600 CE on northern hemisphere cooling via the North Atlantic subpolar gyre and ocean-atmospheric feedbacks. The SPG is hypothesized to be the cooling mechanism that led to low temperature anomalies in Europe and Russia in the early 17th century. While the results are not conclusive, the authors have established a research course to investigate the relationship between volcanic eruptions and important climate shifts that have affected humans. The methods and assumptions of the work are clearly outlined and the authors' interpretation of the results is in accord with their analysis. The supplemental materials make the research reproducible with extensive presentation of the historical observations used in the methods. I found the paper well-structured and written with few technical errors.

The cluster of volcanic activity in the late 1500s would seem to make it difficult to determine if the Huaynaputina eruption seeded the SPG slowdown or if the eruption was the final push over a threshold given the background state of the atmosphere after multiple VEI 4 eruptions. Perhaps this is why an SPG shift can occur in some simulations without a volcanic forcing in 1600. However, the combined use of model simulations, paleoclimate reconstruction, and historical climatology helped to better target an initial seed to the SPG slowdown but unfortunately the data are inconclusive at this point. That said, this is a fine contribution demonstrating how these data sources can be integrated to elucidate the mechanism driving climatic change circa 1600.

We thank the reviewer for the generous comments on the manuscript.  We agree that the current findings are inconclusive regarding the trigger for the possible SPG mechanism.  In the conclusion, we have proposed possible ways the uncertainty could be resolved in future studies, as well as lessons from our experience comparing simulations and data from historical climatology.

Further analysis and discussion of the North Atlantic Oscillation (NAO) and Artic Oscillation (AO) would be helpful to disentangle how the background climate state and internal variability might contribute to an SPG shift. Previous research suggests an interaction between NAO and volcanic eruptions (e.g., Ortega et al. 2015) with a positive NAO emerging after strong volcanic eruptions. NAO+ would lead to stronger westerlies in northern Europe resulting in warmer and wetter winters, meaning the SPG would likely not slowdown. Of course, there are many NAO reconstructions out there to choose from including several recent reconstructions from Ortega et al. 2015, Cook et al. 2019, and Hernandez et al. 2020. The research could benefit from a more comprehensive treatment of NAO/AO.

We thank the reviewer for drawing this issue to our attention.  We propose to expand discussion of the NAO/AO between lines 149 and 150.  As discussed in Hernández et al. 2020, the precise NAO values are uncertain but both the Ortega et al. 2015 and Trouet et al. 2009 studies indicate roughly average NAO index values in the 1590s, declining in the decade following the 1600 eruption.  Thus, the state of the NAO would not appear to be a strong explanation for the cooling before the eruption; nor is there evidence for an NAO+ response following the Huaynaputina eruption, unlike some other tropical eruptions.  As recent studies indicate (Bittner et al., 2016; Coupe and Robock, 2021) a post-eruption NAO+ response with Eurasian winter warming appears to be contingent on tropospheric conditions at the time of the eruption rather than an automatic response to stratospheric aerosols.

Sources:

Bittner M, Schmidt H, Timmreck C, Sienz F. Using a large ensemble of simulations to assess the Northern Hemisphere stratospheric dynamical response to tropical volcanic eruptions and its uncertainty. Geophys Res Lett. 2016;43(17):9324–32

Coupe, J, and Robock, A.: The influence of stratospheric soot and sulfate aerosols on the Northern Hemisphere wintertime atmospheric circulation. J. Geophys. Res. Atmos., 126, e2020JD034513, doi:10.1029/2020JD034513, 2021

Hernández, Armand, Celia Martin-Puertas, Paola Moffa-Sánchez, Eduardo Moreno-Chamarro, Pablo Ortega, Simon Blockley, Kim M. Cobb, et al. "Modes of Climate Variability: Synthesis and Review of Proxy-Based Reconstructions through the Holocene." Earth-Science Reviews 209 (2020): 103286. https://doi.org/10.1016/j.earscirev.2020.103286.

L70 - North Atlantic Oscillation (NAO) was not significantly affected by the eruption but it could be foundation to understanding the background state of climate leading into an SPG shift.

L143-48 – Previous research is showing that no volcanic forcing is need to produce SPG shifts depending on the background climate state. Okay, so what was NAO, or the Artic Oscillation (AO), doing when the SPG shift occurred? How would you distinguish intrinsic variability from a volcanic forcing of an SPG shift?

The role of the background state in the onset of the SPG slowdown was previously discussed in the 2017 Moreno-Chamarro et al. studies. We can also include a plot of the NAO index during 1590–1610 in the different ensembles (that is, with and without volcanic forcing and with and without an SPG shift) to illustrate that neither phase dominates either before or after the Huaynaputina eruption.

It is precisely because we were unable to determine a precise set of initial conditions required for the SPG shift (such as a state of the NAO) that we focused our study on identifying the precise timing of climatic and environmental changes associated with the SPG shift. Although timing alone could not definitely determine whether or not the eruption triggered the SPG shift, it could add strong weight to either inference. If climatic and environmental changes associated with the SPG shift had begun several years after the eruption, as found in the simulations, then we could have concluded that the eruption trigger was more probable *a posteriori*. In fact, we found that those changes -- including increased sea ice and winter cooling -- commenced before the eruption. This finding does not eliminate the possibility of an eruption trigger, since the pre-eruption changes could have arisen due to internal variability or other unidentified forcings. However, it makes an eruption trigger less probable *a posteriori*.

L208 – Is the 1550-1590 baseline period suitable to calculate anomalies when it includes multiple eruptions?

The 1550-1590 baseline excludes the largest eruptions of the 1590s and enables accurate comparison with the simulations (including the no-eruption simulations, which remove eruptions after 1593). The reference period may be extended back another 50 years, but it would not significantly change the results.

L229- the reference period here changes from the reference period for the reconstructed anomalies. Please justify the change in reference period. Or is this a typo?

We thank the reader for identifying this mistake. It was probably a typo, but we will plot the data again using the 1550-1590 reference period to be sure.

L275 – there also appears to be a lack of agreement between NTREND and the simulations. Why might this be?

There is also a high degree of disagreement across the ensemble members, which suggests that internal variability strongly shapes the response to the volcanic eruption. Since the observed historical climate would be similar to another "model realization" -- that is, also strongly shaped by internal variability -- we should not expect perfect agreement between the simulations and observations. To highlight the variety of the model responses to the eruption across the ensemble, we could include a panel in the figure with the temperature anomalies in the ensemble mean of the simulations with volcanic forcing, along with a measure of model agreement.

L282- in Figure 6, it appears that the NVOLC reconstruction has much more annual variability and different spectral properties than the simulations. What is causing this discrepancy?

The blue line represents an average of multiple simulations. Following the strong initial post-eruption response in all simulations, this averaging across simulations reduces variability. To compare the amplitude of variability between the reconstruction and any one simulation, the comparison should be between the black line and the range in the background shading, which are similar.

L297-301 – NAO does play a major role in setting winter conditions in Europe. So, what was the state of the NAO during the period of analysis? L308 – Could the shifts in ice break of dates be connected to NAO and AO? Some of the reconstructions of NAO (Cook 2019, Ortega 2015, Hernandez 2020, etc.) show shifts that might correspond to the ice break up regime shifts.

Please see our previous response addressing these concerns. Although NAO reconstructions leave room for uncertainty, current reconstructions indicate NAO index values in a normal range during the 1590s, with a decline after 1600. We will add discussion of the post-eruption negative NAO anomaly as a possible factor in increased sea ice extent and poor sailing conditions during the first decades of the 17th century (in sections 4.4 and 5.2). Jerjeva (2002) has previously found a negative correlation between the winter NAO index and Baltic sea ice.

Reference:

Jevrejeva, Svetlana. "Association Between Ice Conditions in the Baltic Sea along the Estonian Coast and the North Atlantic Oscillation." Hydrology Research 33, no. 4 (2002): 319–30. https://doi.org/10.2166/nh.2002.0011.

L376-385- If additional simulations do not result in determining what the initial seed for SPG slowdown is, what model improvements would be needed to better model what the climate proxies and historical records appear to show?

If a larger ensemble were insufficient to determine an initial seed, then a higher-resolution model or one that could better represent the volcanic forcing (e.g., with updated

parametrizations or updated inputs for other external forcings such as solar) might also prove necessary.  Furthermore, with higher-resolution reconstructions future studies might specify initial climate conditions in the model world closer to those of the real world.

**Technical Edits**

L31 – add space between number and m - "4,850m"

L140 – missing hypen "Moreno Chamarro"

We thank the review for reading the manuscript closely.  We will make these corrections to the text.

---

## Author Comment (AC5)

**RC 5 with responses**

The study investigates the likelihood of a slowdown in the SPG around the onset of the LIA being linked to the 1600 Huaynaputina volcanic eruption. In order to resolve this issue, the authors attempt to integrate evidence from model-based simulations of past climate conditions with proxy-based paleoclimatic reconstructions and historical records. Despite the inconclusive results, the study highlights both the advantages of adopting an interdisciplinary approach as well as the challenges and limitations of bringing together and interpreting various sources of information.

**General comments:**

In my opinion, the multi-disciplinary nature of this study represents a considerable strength of this work. In general, the manuscript is well written and the findings are presented in a clear and logical manner. The evidence is interpreted objectively and the authors clearly acknowledge the limits of the analysis. From the presented results, conclusions are drawn to the extent that the simulation, reconstruction and limited observational data from the period allow. However, the unconventional structure is rather confusing since the introduction, methods description and some of the results are all blended together, and this also makes it somewhat difficult to distinguish for example what was done in previous studies and what represents original analysis. The authors should therefore seriously consider whether restructuring the manuscript in a more conventional format would be beneficial.

We would like to thank the reviewer for these generous comments on the manuscript. For the most part, the manuscript adopts a conventional structure: Introduction, Methods, Results, Discussion, and Conclusion. The only exception is an additional section (section 2) explaining the modeling and SPG-shift mechanism in the previous 2017 Moreno-Chamarro et al. studies. This additional section prevented the introductory section from becoming overly long and difficult to follow. It also enabled us to explain the previous modeling for a wider audience, including paleoclimatologists and historical climatologists, in keeping with the interdisciplinary scope of the article.

We propose to clarify at the beginning of section 2 that that section explains previous modeling and results, then at the beginning of section 3 that that section introduces the new methods aimed at investigating whether the previously described mechanism could have been triggered by the 1600 Huaynaputina eruption.

Currently, a large part of the discussion is dedicated to discussing the historical / societal impacts of cold conditions at the end of the 16th and during the early 17th century. Greater focus on integrating and discussing the results of the modeling, proxy and historical datasets in more detail would be helpful.

We can expand the discussion in section 5.1 to review the results in greater depth, including the overall agreement of the reconstructions with each other, the relative contribution of each reconstruction to our picture of climate and environmental conditions ca.1600, and the

challenges associated with comparing simulation and reconstruction data (which the reviewer alludes to in the following comment).

Another important point is recognizing and acknowledging discrepancies between model-based simulations with proxy-based reconstructions, which has consequences for understanding uncertainty and the overall reliability of these data sources. This issue is highlighted for example by Figure 6, which shows poor spatial agreement between modeled and reconstructed temperatures. Model simulations are often associated with high uncertainty particularly in relation to post-volcanic cooling and, for example, over-estimation of the magnitude of post-volcanic cooling by some models has been known to occur (e.g. Chylek *et al.*, 2020; Hartl-Meier *et al.*, 2017). Better understanding of some of the shortcomings of these datasets and limitations in their utility within the context of this study could be achieved by exploring a broader set of model simulations or model types to help disentangle the possible influence of model bias and a more detailed examination of the proxy-based temperature reconstructions would also be helpful in this regard.

Some possible sources for discrepancies include: the volcanic forcing itself, which is a simplification and a reconstruction of what happened in the real world; the sensitivity of the model to the forcing; and the impact of internal variability, which might be important enough so that model and data might never be found in agreement for a relatively small ensemble size (which is our case). We will expand section 5.1 to better acknowledge these limitations in the study. However, it would not be within the scope of this study to further test model sensitivity to volcanic forcing.

It is also necessary recognize the potential importance of background climate conditions in modulating the (cooling) response of the North Atlantic to large volcanic eruptions based on the state of the climate system. In relation to this point, the role of internal variability and specifically the potential role of the North Atlantic Oscillation (NAO) in the initiation of SPG weakening and cooler conditions in the north Atlantic sector remains a subject of debate (e.g. Trouet *et al.*, 2009; Lehner *et al.*, 2012). For this reason, some type of examination and discussion of the modes of atmospheric variability in the north Atlantic within this context would be helpful.

We thank the reviewer for drawing this issue to our attention. We propose to expand discussion of the NAO/AO between lines 149 and 150. As discussed in Hernandez et al. (2020) the state of the NAO remains uncertain in this period, but both the Ortega et al. 2015 and Trouet et al. 2009 studies indicate roughly average NAO index values in the 1590s, declining in the decade following the 1600 eruption. Thus, changes in the NAO would not appear to be a strong explanation for the cooling before the eruption; nor do we apparently see the usual post-eruption NAO+ response following Huaynaputina. We can also include a plot of the NAO index during 1590–1610 in the different ensembles (that is, with and without volcanic forcing and with and without an SPG shift) to illustrate that neither phase dominates either before or after the Huaynaputina eruption.

Reference:

Hernández, Armand, Celia Martin-Puertas, Paola Moffa-Sánchez, Eduardo Moreno-Chamarro, Pablo Ortega, Simon Blockley, Kim M. Cobb, et al. "Modes of Climate Variability: Synthesis and Review of Proxy-Based Reconstructions through the Holocene." *Earth-Science Reviews* 209 (2020): 103286. https://doi.org/10.1016/j.earscirev.2020.103286.

One obvious limitation is that most of the presented evidence for the SPG shift is either indirect / circumstantial or entirely model-based. Although the study provides a compelling narrative characterizing anomalously cold conditions in the early 17[th] century, a certain leap of faith is currently required to link an SPG mode shift to these changes. In any case, more information would be required to clarify the relationship between the eruption, short-term and long-term cooling and how these events and changes relate to the state of the SPG. Ultimately, there are limits to the answers that modeling can provide and additional more direct proxy data would likely be required to better understand the dynamics of oceanic circulation and atmospheric dynamics during this period to more precisely pin down the timing, duration and extent of the purported SPG slowdown. Perhaps then it would be possible to confirm or refute the attribution of the observed longer-term cooling in the early 17[th]C, and by extension the initiation of an SPG slowdown, to a volcanic trigger.

The evidence for the SPG shift was provided in the 2017 Moreno-Chamarro et al. studies. This included testing against long-term mainly decadal- to multi-decadal-scale paleoclimate reconstructions. Those reconstructions were insufficient to determine whether the Huaynaputina eruption could have been the trigger for an SPG shift. The new data from high-resolution proxies and historical observations examined in this study enables more precise specification of conditions ca.1600, which raised the possibility of examining whether the previously proposed SPG shift could have been triggered by the Huaynaputina eruption. Additional paleoclimate reconstructions may help determine the possible duration, degree, and extent of an SPG shift; and those additional inferences might help determine whether an eruption triggered an SPG shift in the first place. Nevertheless, such an investigation would be beyond the scope of the current study. The simulations do not currently indicate different types of SPG shifts of different timing, duration, and extent, with some types always triggered by an eruption and others not. Moreover, there is a trade-off between the precise, localized, diverse information provided by historical climatology and the more long-term, continuous, homogenous information provided by paleoclimate reconstructions. Thus, the reconstructions previously used in the 2017 Moreno-Chamarro et al. studies to test the presence of an SPG shift in the real world were less suited for determining whether that shift was triggered by an eruption; while the high-resolution proxies and historical observations used in this study to test for an eruption trigger would be less appropriate for examining the duration or extent of an SPG shift.

**Specific comments:**
L63-72: While it may perhaps be possible for such changes to occur without invoking substantial changes to atmospheric dynamics in the North Atlantic, the background state of the

atmosphere, internal variability and the role of the NAO cannot be discounted *a priori*, particularly as these factors may act to modulate the response of the climate system to a large volcanic event.

(Please see the above response to concerns about the NAO state at the time of the eruption.)

L89: The phrase 'possibilities for adaptation' seems a bit vague and it is not clear what this refers to. Please specify / clarify this point.

We could clarify the phrase "nature of societal vulnerabilities and possibilities for adaptation" as "which activities and institutions were vulnerable, and how people could adapt them to changing climatic and environmental conditions".

Figure 2: For easier interpretation of the figure, it may be clearer to also state in the panel sub-headings that the plots are showing temp. / Sv. anomalies.

We thank the reviewer for the suggestion and will correct the figures accordingly.

L233: It is not clear whether this implies that only a 30-yr segment length was used or a range of segment lengths (30-yr+) was examined. If it is the former case, please remove 'minimum' to avoid confusion. Otherwise, please specify the range of segment lengths utilized.

The segment length (minimum number of observations between the changes) in the change point analysis is ≥ 30-years. We will change the language for clarification.

Figure 5: Please specify in the figure caption what the purple dots in top-left plot represent.

The purple dots in top-left plot of Figure 5 are the tree-ring width and maximum latewood density sites that have been used for the spatial reconstruction

L203-210: What was the size of the reconstructed grid cells? Which instrumental dataset was used for calibration? How were the chronologies merged and how was the reconstruction performed (e.g. PCA, nesting), etc.? In general, more detailed information about the development of the spatial reconstruction is needed here (or at least in supplementary materials).

The size of reconstructed grid cells is 5x5° lat/long. The instrumental data used as target field for the reconstruction are May to August monthly temperature anomalies wrt the 1961-1990 period extracted from the HadCruT4 (Cowtan and Way, 2014). The spatial reconstruction was developed using a point-by-point regression (Cook et al., 1999) which accounts for the spatial distribution and relationship of the proxy predictor network to the target field.

References:

Cook, Edward R., David M. Meko, David W. Stahle, and Malcolm K. Cleaveland. "Drought Reconstructions for the Continental United States." Journal of Climate 12 (1999): 1145–62. https://doi.org/10.1175/1520-0442(1999)012<1145:DRFTCU>2.0.CO

Cowtan, Kevin, and Robert G. Way. "Coverage Bias in the HadCRUT4 Temperature Series and Its Impact on Recent Temperature Trends." Quarterly Journal of the Royal Meteorological Society 140, no. 683 (2014): 1935–44. https://doi.org/10.1002/qj.2297.

Figure 6: How does the NVOLC reconstruction compare with N-TREND (and model output) over the investigated period? Currently, only NVOLC is compared to model output, whereas N-TREND is only used for illustration and is not compared to NVOLC or the modeled temperatures. The highly anomalous cooling in SE Europe in the NVOLC reconstruction (Fig. 6a) is rather suspicious and I wonder how robust this feature is. According to Supplementary Figure S3 in Guillet et al. (2017), most of northern Europe and parts of western / southwest Europe calibrate well, whereas calibration / verification statistics are very weak for NW, central and especially eastern and SW Europe. Consider that poor spatial representation of reconstructed temperatures may cause disagreement with modeled temperatures in some areas. Likewise, specific limitations of the model may also lead to disagreement. Such considerations should be acknowledged and discussed.

We will include additional discussion of the limitations of spatial representation of temperature anomalies in the NVOLC.  Comparison with the N-TREND reconstruction may create confusion, however, since N-TREND includes many more TRW chronologies (with the problem of memory in series) than the NVOLC dataset, which was created with the goal to detect and quantify volcanic cooling and thus includes temperature-sensitive MXD records.

L286: Why is the NVOLC v2 reconstruction shifted by +0.5 K?

The aim of the figure is to compare the forcing generated by the 1600 eruption in the simulations and reconstruction, rather than absolute temperatures.

L350: I suggest that a more appropriate term to use in this context would be 'support' rather than 'appear to confirm'.

We thank the reviewer for the suggestion and will correct the sentence accordingly.

L368-370: So, considering the timing, might this in fact suggest that the Huaynaputina eruption is rather unlikely to be the cause of the SPG slowdown?

It remains difficult to say.  As we explain in the next paragraph of the manuscript, the results would be consistent with a scenario in which the eruption did trigger an SPG shift but colder conditions had already started during the 1590s due to internal variability or a different forcing.  Therefore, we could say that the results do *not confirm* a scenario in which the 1600 eruption triggered an SPG shift, but we couldn't say that the results *contradict* such a scenario.  In terms of inference, our findings should reduce posterior estimates of the

probability for an eruption trigger (per Bayes' theorem) since the likelihood of our reconstruction data given an eruption trigger is lower than the likelihood of getting that data regardless of the eruption trigger.  Whether the eruption trigger hypothesis is *a posteriori* improbable -- i.e., $p(h|d) < 0.5$ -- would depend on one's prior probability estimate for the hypothesis.

L371-385: Another possibility could be that a pronounced shorter-term cooling impact of the Huaynaputina eruption was 'superimposed' on the longer-term cooling trend, which may have been initiated prior to 1600 (either in response to the cluster of late-16th century volcanic eruptions or otherwise). Evidence for volcanic-induced short-term cooling is on firmer ground as the results are consistent with this type of response to the eruption (Fig. 6c) and this is also consistent with the duration and magnitude of inferred NH cooling responses to large (tropical) eruptions more generally based on proxy reconstructions (e.g. Esper et al. 2015) and modeling of surface air temperature. In contrast, the mechanism for initiating longer-term cooling / SPG slowdown and attribution of such changes to a particular volcanic event is highly uncertain and rather problematic.

We thank the reviewer for this suggestion.  It is, of course, possible that none of the simulations have captured the mechanism for cooling found in the real world, and we will note this at the end of the paragraph on line 375.  However, as previously discussed in the 2017 studies by Moreno-Chamarro et al., 8 of the 20 simulations (6/10 with volcanic forcing and 2/10 without) reproduced an SPG shift as well as long-term winter cooling and increased sea ice extent found in previous reconstructions.  Thus, we have chosen to investigate this mechanism more closely and to assess different scenarios indicated by those simulations.

L376-377: One could argue that it is uncertain whether this issue could be definitively resolved through modelling alone.

We will revise the sentence to read "First, *further comparison between high-resolution reconstructions and* a larger ensemble of climate simulations could improve…"

L395-397: I would recommend reformulating this sentence considering that, based on extensive paleoclimatic evidence, the occurrence of cool (wet) summers during this period is actually not in question. Therefore, rather than 'confirming' this, it would be more appropriate to state that this study provides further support and a broader context for such conditions at that time.

We thank the reader for this suggestion and will revise the sentence accordingly.

**Minor / technical comments:**
L95: consider '… activation (or lack thereof) …'

L186/380/384: 'an SPG' rather than 'a SPG'

Fig.4 legend / L233 / (L303): 'ice break-up timing data' instead of 'date data'

L223: consider 'obtained' instead of 'gained'

L238: consider 'recorded' rather than 'left'

L239: 'latter' rather than 'later'?

L243: Suggested wording adjustment: 'These observations were recorded in areas with flat terrain …'

L266: 'did or did not turn back' or alternatively 'ships could pass or were forced to turn back'

L267: 'turn back during a voyage' or perhaps 'terminate a voyage'?

L268: 'cold conditions' or 'cold temperatures' / 'dangerous sailing conditions' or 'danger posed by sailing conditions'

L280: Change 'NTREND' to 'N-TREND'. Also, something is missing here - consider: '… in each year over the 1601-1609 period …'?

L287: should the range be 1593-1650 instead of 1593-1640?
Yes. This apparently a typo.

L290: Consider: 'The analysis in Figure 7 indicating the ice …'

L293: 'can occur' rather than 'can happen'

L303: please change 'wrt' to 'w.r.t.'

L308: 'detected in' rather than 'detected at'?

L362: 'as a Possible Trigger'?

L372: remove 'has'?

L375: 'any role of'?

We thank the review for reading the manuscript closely. We will correct these typos and clarify the phrasing in the lines indicated above.

---

## Author Response (AR1)

**Responses to Review Comments and Explanation of Changes**

**RC1**

The lead author is one of the world's key scholars in Historical Climatology and Climate History cooperating with natural scientists and historians, mainly from Europe. The Huaynaputina is a stratovolcano in southern Peru. Its eruption in February 1600 CE triggered a persistent summer and winter cooling in the North Atlantic region during the early 17. It-was the largest eruption in the Andes in historical times. The paper explores the eruption-induced cooling mechanism in some detail. The study compares simulations and a North Atlantic subpolar gyre shift in annual proxies in archives of nature to more detailed with documentary evidence.

The Huyaputina injected fewer sulfates into the stratosphere than other LIA eruptions with a compatible cooling effect. In particular the eruption generated much more and longer lasting winter cooling than summer cooling in Central and Northern Europe, which is unusual. It is hypothesized that reduced heat transport by the North Atlantic Subpolar Gyre SPG in terms of a cooling of the North Atlantic may be one of the reasons, though many aspects remain unclear. Historical written records as well as contemporary historical observations of relevant climate and environmental conditions demonstrate patterns of cooling and sea ice expansion consistent with, but not necessarily indicative of an eruption trigger for the proposed SPG slowdown mechanism

The arguments of scientists and modellers should still be improved in view of the limited understanding of people from the historical sciences for processes in the ocean. Some relevant studies might still be included: The herring catch on the west coast of Scotland declined remarkably between 1585 and 1597 which Parry (1978) interpreted this as an escape of this cold-sensitive species from the cold water masses advancing southwards. The detailed temperature reconstruction by Dobrovolny et al. (2010) for Western and Central Europe since 1500 CE was overlooked. It contains seasonal temperature that are explained in more detail by the recent synthesis by Pfister and Wanner (2021) that is published on 6[th] September. These data show that the very severe winter of 1600 preceded the Huynaputina eruption in contrast to the cold winter in 1601. Likewise, just spring 1600 was very cold. On the other hand, the summer 1601 was very cold.

We thank the author for his generous comments on the manuscript. The fish catch data in Parry 1978—as well as Jürgen Alheit and Eberhard Hagen, "Long-Term Climate Forcing of European Herring and Sardine Populations," *Fisheries Oceanography* 6 (1997): 130–39—indicate declining herring catch at the north end of their range and expansion of the south end of their range by the 1590s, which would be consistent with our finding of cooling preceding the 1600 eruption; although as discussed in Poul Holm et al. "The North Atlantic Fish Revolution (ca. AD 1500)." *Quaternary Research*, 2019, larger shifts in the fishing industry, including the abundance of previously unexploited fisheries being discovered off North America, make it difficult to attribute fluctuations in European fish catch during the late 1500s solely to climatic changes. The Moreno-Chamarro et al. 2017 studies presenting the SPG shift mechanism already considered the Dobrovolny et al. 2010 reconstructions (as well the Luterbacher et al. 2004 reconstruction).

We have added the following sentences to the manuscript before line 90: "Moreover, some historical climatology evidence indicates cooling in Europe and the North Atlantic prior to

the 1600 eruption. Winter temperature reconstructions of Central Europe based on contemporary written observations indicate an onset of colder winters by the 1590s (Dobrovolný et al., 2010; Pfister and Wanner, 2021). Furthermore, historic fishing records indicate an expanded herring catch at the southern end of the herring range during the 1590s, which would also be consistent with cooling North Atlantic temperatures prior to 1600, although this shift may have alternative explanations (Alheit and Hagen, 1997; Holm et al., 2019). Therefore, further higher-resolution reconstructions and examination of historical observations are required to determine whether or not such an SPG shift was triggered by the Huaynaputina eruption."

**RC2 with responses**

The manuscript fits the scope of the journal well with integration of climate modeling, paleoclimatology, and historical climatology to address the effect of the Huaynaputina (Peru) eruption in 1600 CE on northern hemisphere cooling via the North Atlantic subpolar gyre and ocean-atmospheric feedbacks. The SPG is hypothesized to be the cooling mechanism that led to low temperature anomalies in Europe and Russia in the early 17[th] century. While the results are not conclusive, the authors have established a research course to investigate the relationship between volcanic eruptions and important climate shifts that have affected humans. The methods and assumptions of the work are clearly outlined and the authors' interpretation of the results is in accord with their analysis. The supplemental materials make the research reproducible with extensive presentation of the historical observations used in the methods. I found the paper well-structured and written with few technical errors.

The cluster of volcanic activity in the late 1500s would seem to make it difficult to determine if the Huaynaputina eruption seeded the SPG slowdown or if the eruption was the final push over a threshold given the background state of the atmosphere after multiple VEI 4 eruptions. Perhaps this is why an SPG shift can occur in some simulations without a volcanic forcing in 1600. However, the combined use of model simulations, paleoclimate reconstruction, and historical climatology helped to better target an initial seed to the SPG slowdown but unfortunately the data are inconclusive at this point. That said, this is a fine contribution demonstrating how these data sources can be integrated to elucidate the mechanism driving climatic change circa 1600.

We thank the reviewer for the generous comments on the manuscript. We agree that the current findings are inconclusive regarding the trigger for the possible SPG mechanism. In section 5.1 and the conclusion, we have proposed possible ways the uncertainty could be resolved in future studies, as well as lessons from our experience comparing simulations and data from historical climatology.

Further analysis and discussion of the North Atlantic Oscillation (NAO) and Artic Oscillation (AO) would be helpful to disentangle how the background climate state and internal variability might contribute to an SPG shift. Previous research suggests an interaction between NAO and volcanic eruptions (e.g., Ortega et al. 2015) with a positive NAO emerging after strong volcanic eruptions. NAO+ would lead to stronger westerlies in northern Europe resulting in warmer and wetter winters, meaning the SPG would likely not slowdown. Of course, there are many NAO reconstructions out there to choose from including several recent reconstructions from Ortega et al. 2015, Cook et al. 2019, and Hernandez et al. 2020. The research could benefit from a more comprehensive treatment of NAO/AO.

L70 - North Atlantic Oscillation (NAO) was not significantly affected by the eruption but it could be foundation to understanding the background state of climate leading into an SPG shift.

L143-48 – Previous research is showing that no volcanic forcing is need to produce SPG shifts depending on the background climate state. Okay, so what was NAO, or the Artic Oscillation (AO), doing when the SPG shift occurred? How would you distinguish intrinsic variability from a volcanic forcing of an SPG shift?

We thank the reviewer for drawing this issue to our attention.  The role of the background state in the onset of the SPG slowdown was previously discussed in the 2017 Moreno-Chamarro et al. studies.   As discussed in Hernández et al. 2020, the precise NAO values are uncertain but both the Ortega et al. 2015 and Trouet et al. 2009 studies indicate roughly average NAO index values in the 1590s, declining in the decade following the 1600 eruption.  Thus, the state of the NAO would not appear to be a strong explanation for the cooling before the eruption; nor is there evidence for an NAO+ response following the Huaynaputina eruption, unlike some other tropical eruptions.  As recent studies indicate (Bittner et al., 2016; Coupe and Robock, 2021) a post-eruption NAO+ response with Eurasian winter warming appears to be contingent on tropospheric conditions at the time of the eruption rather than an automatic response to stratospheric aerosols.

We have added the following to the manuscript at line 169: "Nor did we find that the onset of the SPG shift depended on a particular state of the NAO. Neither simulations with an SPG nor those without display consistent or anomalous high or low NAO index values in the decades before or after 1600, as shown in Figure 2. Furthermore, different reconstructions indicate different NAO index values during this period (**Figure 2** and Hernandez et al., 2020), but neither the reconstructions nor simulations display a positive NAO anomaly after 1600 such as those identified following other large tropical eruptions (e.g., Christiansen, 2008). As recent studies indicate (Bittner et al., 2016; Coupe and Robock, 2021), a post-eruption positive NAO response with Eurasian winter warming appears to be contingent on tropospheric conditions at the time of the eruption rather than a dynamical response to stratospheric aerosols alone."  The new figure 2 illustrates NAO index reconstructions and values in simulations with and without volcanic forcing and SPG shift.

It is precisely because we were unable to determine a precise set of initial conditions required for the SPG shift (such as a state of the NAO) that we focused our study on identifying the precise timing of climatic and environmental changes associated with the SPG shift.  Although timing alone could not definitely determine whether or not the eruption triggered the SPG shift, it could add strong weight to either inference.  If climatic and environmental changes associated with the SPG shift had begun several years after the eruption, as found in the simulations, then we could have concluded that the eruption trigger was more probable *a posteriori*.  In fact, we found that those changes -- including increased sea ice and winter cooling -- commenced before the eruption.  This finding does not eliminate the possibility of an eruption trigger, since the pre-eruption changes could have arisen due to internal variability or other unidentified forcings.  However, it makes an eruption trigger less probable *a posteriori*.

Sources:

Bittner M, Schmidt H, Timmreck C, Sienz F. Using a large ensemble of simulations to assess the Northern Hemisphere stratospheric dynamical response to tropical volcanic eruptions and its uncertainty. Geophys Res Lett. 2016;43(17):9324–32

Coupe, J, and Robock, A.: The influence of stratospheric soot and sulfate aerosols on the Northern Hemisphere wintertime atmospheric circulation. J. Geophys. Res. Atmos., 126, e2020JD034513, doi:10.1029/2020JD034513, 2021

Hernández, Armand, Celia Martin-Puertas, Paola Moffa-Sánchez, Eduardo Moreno-Chamarro, Pablo Ortega, Simon Blockley, Kim M. Cobb, et al. "Modes of Climate Variability: Synthesis and Review of Proxy-Based Reconstructions through the Holocene." Earth-Science Reviews 209 (2020): 103286. https://doi.org/10.1016/j.earscirev.2020.103286.

L208 – Is the 1550-1590 baseline period suitable to calculate anomalies when it includes multiple eruptions?

The 1550-1590 baseline excludes the largest eruptions of the 1590s and enables accurate comparison with the simulations (including the no-eruption simulations, which remove eruptions after 1593).  The reference period may be extended back another 50 years, but it would not significantly change the results.

L229- the reference period here changes from the reference period for the reconstructed anomalies. Please justify the change in reference period. Or is this a typo?

We thank the reader for identifying this mistake.  This was a typo, and we have plotted the data again using the 1550-1590 reference period to be sure.

L275 – there also appears to be a lack of agreement between NTREND and the simulations. Why might this be?

We directly compare the simulations only to the NVOLC reconstruction, in figure 7 (previously figure 6).  There is also a high degree of disagreement across the ensemble members, which suggests that internal variability strongly shapes the response to the volcanic eruption. Since the observed historical climate would be similar to another "model realization" -- that is, also strongly shaped by internal variability -- we should not expect perfect agreement between the simulations and observations. To highlight the variety of the model responses to the eruption across the ensemble, we have updated figure 7 to include separate panels for (b) the closest simulation to the reconstruction; (c) the average for simulations with volcanic forcing; (d) the average for simulations without volcanic forcing.

We have also expanded the discussion at line 409 to consider reasons for reconstruction-simulation discrepancies: "The large degree of short-term summer cooling in the NVOLC v2 reconstruction is similar to the mean of simulations with an SPG shift. Nevertheless, the spatial pattern of the anomaly differs between the reconstruction and those simulations. There are several possible sources for this discrepancy.  The NVOLC v2 reconstruction has weaker spatial coverage in southeastern Europe, where some of the strongest differences appear (Guillet et al., 2017). The climate model necessarily simplifies climatic processes. Most importantly, this study has considered a relatively small ensemble size (10 realizations with volcanic forcing and 10 without) within a single model, as well as a single reconstructed history of volcanic forcing. Beyond limitations due to the specificity of the chosen model and forcing, the ensemble size seems to be insufficient to encompass the range of possible climate responses to the Huaynaputina eruption that stems from their dependency on the initial state of the climate system at the time of the eruption."

L282- in Figure 6, it appears that the NVOLC reconstruction has much more annual variability and different spectral properties than the simulations. What is causing this discrepancy?

The blue line represents an average of multiple simulations. Following the strong initial post-eruption response in all simulations, this averaging across simulations reduces variability. To compare the amplitude of variability between the reconstruction and any one simulation, the comparison should be between the black line and the range in the background shading, which are similar.

L297-301 – NAO does play a major role in setting winter conditions in Europe. So, what was the state of the NAO during the period of analysis? L308 – Could the shifts in ice break of dates be connected to NAO and AO? Some of the reconstructions of NAO (Cook 2019, Ortega 2015, Hernandez 2020, etc.) show shifts that might correspond to the ice break up regime shifts.

Please see our previous response addressing these concerns. Although NAO reconstructions leave room for uncertainty, current reconstructions indicate NAO index values in a normal range during the 1590s, with a decline after 1600. We have expanded the discussion of ice break-up dates (section 4.2) to include the following: "This result also confirms that intrinsic climate variability, including climate modes such as the NAO and Arctic Oscillation, plays a major role in setting winter conditions over Europe, including Baltic Sea temperatures and sea ice (Moreno-Chamarro et al., 2017b; see also Chen and Hellström, 1999; Omstedt and Chen, 2001; Eriksson et al., 2007; Zanchettin et al., 2019). In particular, the duration of Baltic sea ice and port closures shows a negative correlation with the NAO index (Jevrejeva, 2002); thus, the absence of a post-eruption positive NAO anomaly helps account for the longer port closure dates ca. 1600."

Reference:

Jevrejeva, Svetlana. "Association Between Ice Conditions in the Baltic Sea along the Estonian Coast and the North Atlantic Oscillation." Hydrology Research 33, no. 4 (2002): 319–30. https://doi.org/10.2166/nh.2002.0011.

L376-385- If additional simulations do not result in determining what the initial seed for SPG slowdown is, what model improvements would be needed to better model what the climate proxies and historical records appear to show?

We have expanded discussion section 5.1 to discuss the following possibilities: "Our results, although inconclusive, suggest two ways forward. First, further comparison between high-resolution reconstructions and a larger ensemble of climate simulations could improve the chances for determining whether or not the 1600 Huaynaputina eruption triggered the SPG slowdown. With a larger ensemble, it may become possible to identify an initial seed for the SPG slowdown mechanism, since the signal-to-noise ratio of emergent features increases with the ensemble size. Identification of such an initial seed in proxy reconstructions and historical observations may enable more certain identification of an SPG slowdown and its causes than the attempts to find characteristic effects of such a slowdown in this study. This type of study may require large ensembles of simulations (20 members or more) in order to detect a clear signal for the onset and evolution of a climate mechanism above the noise of interannual climate variability. If a larger ensemble were insufficient to determine an initial

seed, then a higher-resolution model might also prove necessary, or else a model that could better represent the volcanic forcing (e.g., with updated aerosol parametrizations or updated volcanic forcing histories) and other external forcings (such as solar). Furthermore, with higher-resolution reconstructions, future studies might specify initial climate conditions in the model world closer to those of the real world.

Second, additional high-resolution climate proxies and historical records covering more locations in the North Atlantic could help determine whether anomalies during the late 1590s were indicative of an SPG slowdown preceding the 1600 Huaynaputina eruption. Moreover, forward modeling might be used to directly simulate additional proxies and conditions present to contemporary observers in order to provide stronger tests of an eruption trigger and SPG shift using historical climatology data. However, such modeling may have to take into account not only physical processes but human processes of observation, recording, and transmission."

**Technical Edits**

L31 – add space between number and m - "4,850m"

L140 – missing hypen "Moreno Chamarro"

We thank the review for reading the manuscript closely. We have made these corrections to the text.

**RC 3 with responses**

Summary: The focus of the study is to disentangle the role of the external volcanic forcing and of internal climate variability in the cooling surrounding the strong volcanic eruption in of the Huaynaputina eruption 1601. The authors analyse different proxy data sets, including air,temperature, sea-ice extent in the Arctic-North-Atlantic, historical ice break-up dates at several Baltic ports and ensemble of simulations with the Max-Plank-Institute Earth System model.

The conclusions is that the attribution of all aspects of climate change around this date are very difficult to disentangle. Internal climate variability may be large, and although the attribution of the whole temperature evolution around those decades is compatible with the effect of volcanic eruptions, internal processes may play also an important role, even before the eruption.

Recommendation: I enjoyed reading this paper very much, and I like to congratulate the authors at this point. Although, as the authors acknowledge, the study is not conclusive, the authors have tried to use all data sets available to them and have conducted a very thorough, objective and candid analysis. On the other hand, it is very well written, provides an exhaustive background on the physical mechanisms and on the historical evidence, also a proof of a well functioning collaboration between climatologist and historians. Perhaps the significance of this and similar studies goes beyond what the authors let on: this type of events can occur at any time, regardless of whether they are produce only by volcanic forcing or by a combination of volcanic eruption and internal variability. This, it is important to understand these past events.

This manuscript is one of the best that I had a chance to evaluate. My recommendation is to publish it - I have just a few minor comments that the authors may want to consider.

1) My most general comment is directed to the quantification of the magnitude of internal climate variations in this context. The study does compare simulations with and without external volcanic forcing for this period, but here a more general question remains open: what is the largest magnitude of multi-year cooling in a long control simulation in this region? Do large periods of multiyear cooling, comparable to the years following the 1601 eruptions, also appear in simulations without variations in external forcing?

We thank the reviewer for the generous comments on the manuscript. Moreno-Chamarro et al. 2015, "Internally generated decadal cold events in the northern North Atlantic and their possible implications for the demise of the Norse settlements in Greenland," has previously addressed this question. The largest cooling in the subpolar North Atlantic found in a control simulation with constant forcing simulations and without volcanic forcing is found in those ensembles with an SPG shift, where the has a magnitude of cooling is up to -2˚C for 30-40 years

2) line 34 'The VEI 6 1600 Huaynaputina eruption' I think VEI has not been defined at this point in the manuscript.

We have revised the sentence to define volcanic explosivity index (VEI).

3) line 145 ' Of the 8 total members of the SPG-shift ensemble, 6 had volcanic forcing' . How many simulations with volcanic forcing do not show a SPG-shift ?

There were 10 runs in the VOL ensemble (6 with shift, 4 without) and 10 runs in the NOVOL ensemble (2 with shift, 8 without). We have revised lines 159-160 to include this information.

**RC 4 with responses**

**General**
The paper investigates the climatic and historical context of a tropical volcanic eruption at the beginning of the 17th century using a combination of proxy, historical and modeling evidence. The authors initially hypothesize a prominent influence of the state of the North Atlantic Subpolar Gyre (SPG), initiating a sustained cooling over Europe. Authors conclude that the SPG could have an important role. However, their results remain inconclusive taking into account their combined evidence using different reconstructed metrics (e.g. Baltic Sea Ice, North Sea Winds).

The SPG-shift hypothesis came from the previous studies by Moreno-Chamarro et al. (2017a, 2017b). It is not an original hypothesis of this manuscript; nor does this manuscript set out to prove the existence of this mechanism. Instead, building on past studies supporting the SPG-shift mechanism and using the new data drawn from high-resolution proxies and historical observations, we aim to (1) specify climatic and environmental conditions ca.1600 in order to (2) determine whether such an SPG shift could have been triggered by the 1600 Huaynaputina eruption and thereby to (3) evaluate how to compare data from historical climatology (e.g., written descriptions and proxies from human activities) with simulations. It appears that most of the referee's subsequent concerns and recommendations derive from this misimpression of the manuscript's scope and intentions.
We have added clarifications in several places to avoid this misimpression among future readers, including the following:

We have specified that the study compares *previous* modeling with new reconstructions through the abstract and sections 1 and 2.
The beginning of section 2 now reads: "This section reviews the previous modeling results that established the SPG mechanism for persistent cooling and its consistency with previous paleoclimate reconstructions, as well as the challenges in determining whether or not this mechanism was triggered by the 1600 Huaynaputina eruption."
The beginning of section 3 now reads: "This section presents the new high-resolution paleoclimate proxies and historical observations of climate and environmental conditions selected to determine whether these were consistent or not with a Huaynaputina eruption trigger for the previously identified SPG mechanism for persistent cooling."

The basic structure of the manuscript is not in an optimal shape. The reader is confronted with different parts of a classical paper, resulting in a mix of background information, methods, data sets and conclusions presented in the different sub-chapters. This might be sourced in the fact that it is an interdisciplinary research paper. However, some crucial statements and physical mechanisms contradict in different sections of the paper. Therefore I suggest that i) the manuscript should be completely re-written ii) the proxy- and historically derived hypotheses should be clearly formulated in the beginning and (statistically) tested by a more comprehensive suite of available CMIP6 simulations [ e.g. those CMIP6 model simulations with appropriate ocean models and according horizontal and vertical resolution – in the present manuscript only one Earth System Model is used ] and iii) the different parts and disciplines should be conceptually better coordinated.

The SPG-shift hypothesis was not derived from the high-resolution proxies and historical observations examined in this study.  The SPG mechanism was derived from modeling in previous studies and tested against other longer-term, lower-resolution reconstructions in previous studies (Moreno-Chamarro 2017a, 2017b).  Given their greater spatial coverage, homogeneity, and continuity, the longer-term, lower-resolution reconstructions considered in those previous studies were more suited to testing the hypothesis that an SPG shift such as that found in the simulations was the cause of persistent winter cooling in the real world.  However, those reconstructions were unable to test whether the 1600 eruption was the trigger for such an SPG shift.  Although such an SPG shift occurred more often in simulations with eruptions, it also occurred in 2 simulations without eruptions.
Thus, previous studies left open an interesting question: If the SPG-shift mechanism did occur, was it triggered by the 1600 Huaynaputina eruption?
Moreno-Chamarro et al. 2017a, 2017b tried to find an answer to this question through sensitivity experiments in the model world, but they did not reach a definite conclusion.  Our present study has therefore drawn on new high-resolution proxies and historical observations to address this question.  We first attempted to identify within the simulations some necessary set of initial conditions for the SPG shift that might be tested in those observations.  However, no such initial seed could be identified.  Because we were unable to determine some testable initial seed for the SPG shift (such as a state of the NAO), we focused our study on identifying the precise timing of climatic and environmental changes associated with the SPG shift.  Although timing alone could not definitely determine whether or not the eruption triggered the SPG shift, it could add strong weight to either inference.  If climatic and environmental changes associated with the SPG shift had begun several years after the eruption, as found in the simulations, then we could have concluded that the eruption trigger was substantially more probable *a posteriori*.  In fact, we found that those changes -- including increased sea ice and winter cooling -- commenced before the eruption.  This finding does not eliminate the possibility of an eruption trigger, since the preeruption changes could have arisen due to internal variability or other unidentified forcings. However, it makes an eruption trigger less probable *a posteriori*. This investigation has also enabled us to better explain the societal impacts of climatic variability and change ca.1600 and to make recommendations for future comparisons of simulations and data from historical climatology.

Within this scope, the manuscript adopts a standard structure (Introduction, Methods, Results, Discussion, Conclusion). The only exception is an additional section explaining the modeling and SPG-shift mechanism in the previous Moreno-Chamarro 2017 studies (section 2). This additional section prevented the introductory section from becoming overly long and difficult to follow. It also enabled us to explain the previous modeling for a wider audience, including paleoclimatologists and historical climatologists, in keeping with the interdisciplinary scope of the article. It was not intended as a substitute for a "methods" section about modeling. No new modeling was performed for the present manuscript, and any such modeling would be beyond the means and scope of this study.

In the following I provide some suggestions how to re-structure the manuscript.

**Specific**

*Introduction/Basic Concept*

The introduction should contain the basic background information to understand the concept and eventually the conclusions of the paper. Therefore it is of ultimate importance to be conceptually sound and also introduce the main concepts elaborated in the manuscript. For instance, the SPG is not introduced at all. Also, the model used is never explained related to the fact how the model is capable to realistically simulate the SPG. For this a dedicated methods and data section is necessary where the different components of the paper are comprehensively explained. In the current version the first part is a mere repetition of results already published elsewhere (Morene-Camaro et al., 2017) even including the same set of figures (cf. Figure 1) that is already published.

There was no new modeling performed for this study; and additional modeling would be beyond its scope. For the modeling methods, please see the 2017 Moreno-Chamarro et al. studies. The methods in this study concern the reconstruction of climate and environmental conditions from high-resolution proxies and historical observations and their comparison to previously developed simulations.

This also relates to the 2nd important concern: The authors only use one model from the CMIP6 suite. For their period under consideration a larger number of simulations, also for ocean models, is available in the Earth System Grid Federation (ESGF) platform. Since results for MPI already have been published and this very mechanism might be evident only in the MPI model, the question is whether all of the CMIP6 models, or at least those with similar horizontal resolution show a similar response. This is even more important when comparing model derived results to real-world derived hypotheses to test the robustness of the model derived results and to finally discriminate between I) internal vs. forced changes and ii) model-vs-model intrinsic variability related to the individual structure of the Earth System Model.

This study has drawn on new high-resolution proxies and historical observations to specify conditions around the time of the 1600 Huaynaputina eruption and thereby determine whether that eruption could have triggered an SPG shift similar to that found in previous simulations. There was no new modeling performed for this study, and additional modeling to run tests of the SPG mechanisms would be beyond its scope.

The previous studies (Moreno-Chamarro 2017a, 2017b) used data only from the MPI-ESM-P model (in its CMIP5 configuration) because it is the only one for which we had a dedicated set of sensitivity simulations to explore the impact of the Huaynaputina eruption on the climate. Including other last millennium simulations would have blurred the discussion because it would have meant dealing with different models and model sensitivity to external forcings, different volcanic forcings, different background climate states at the time of the eruption, and different internal variability. The only model with a close enough setup is the CESM last millennium ensemble, which includes sensitivity simulations with different external forcing. However, it is not clear whether this model, the CESM-CAM5_CN, shows any sensitivity in the subpolar region to the volcanic forcing (Otto-Bliesner et al., 2016), although a newer model version shows cooling in the North Atlantic during the Little Ice Age associated with a SPG weakening (Zhong et al., 2018). There is currently limited data for other CMIP6/PMIP4 last millennium simulations available at the ESGF nodes (and not for the MPI-ESM model).

References:
Otto-Bliesner, B.L., Brady, E.C., Fasullo, J., Jahn, A., Landrum, L., Stevenson, S., Rosenbloom, N., Mai, A. and Strand, G., 2016. Climate variability and change since 850 CE: An ensemble approach with the Community Earth System Model. Bulletin of the American Meteorological Society, 97(5), pp.735-754
Zhong, Y., Jahn, A., Miller, G.H. and Geirsdottir, A., 2018. Asymmetric Cooling of the Atlantic and Pacific Arctic During the Past Two Millennia: A Dual Observation-Modeling Study. Geophysical Research Letters, 45(22), pp.12-497.)

*Methods/Hypotheses*
The basic setup of the authors using a combination of different disciplines to address a certain questions is a good asset. However, the potential and per-requisites should be formulated conceptually more sound. For instance, in the present setup the hypotheses should be derived based on proxy and/or historical evidence. In a second step a potential physical mechanism should be motivated explaining the initially formulated hypothesis (e.g. changes in SPG and its impact on European temperatures). In a third step this should then be tested in the model world, most preferentially using a suite of comprehensive Earth System Models simulating this period and using state-of-the art statistical tests (for instance Boot Strap methods using control simulations to derive reference climatic states). In the present version this concept is reversed and the initial hypotheses are derived from the climate model. In general, this is also possible but it is of ultimate importance to state this clearly and also present a way of how this (set of) hypotheses is falsified.

The formulation and testing of a new climate mechanism is beyond the scope of this study. The 2017 Moreno-Chamarro et al. studies already developed and tested the mechanism examined in this study. Our primary question is: If the SPG-shift mechanism examined in those previous studies did occur, could it have been triggered by the 1600 Huaynaputina eruption? To answer that question, we have used high-resolution proxies and historical observations best suited for specifying local and regional conditions ca.1600.

The lack of a sound statistical testing scheme and a careful inspection of the different conclusions derived in specific parts of the manuscript results e.g. in the following contradicting statement:
*Moreno Chamarro et al. (2017b) found consistencies at multidecadal scales between simulations with a weakened SPG and reconstructed changes in several geophysical*

*variables of the North Atlantic after ca. 1600 CE. The study did not conclude that the late 16 th -century volcanic cluster was necessary for the SPG shift, which was instead mainly attributed to intrinsic variability of the simulated climate system. Sensitivity simulations of the period 1593-1650 with no volcanic forcing yielded SPG shifts similar to those in the volcanically forced simulations. [ cf l. 141 ff ]*
*vs.*

1. *This study has examined high-resolution proxies and historical observations to investigate whether the 1600 Huaynaputina eruption triggered persistent cooling in the North Atlantic region by initiating a regime-shift of the North Atlantic subpolar gyre toward a persistent weak phase in the early 17 th century,* **as shown by paleoclimate model simulations**. *[ cf. l. 371 ff. ]*

The conclusion derived from the model analysis showed that the shift might be simply due to intrinsic or internal climate variability. However, in the conclusions authors sate the volcanic eruption triggered the regime shift initiated by the volcanic eruption. Moreover, the paleoclimatic model simulations only relate to the MPI-ESM model the authors used for their investigations.

The "as shown by paleoclimate model simulations" referred to the second part of that clause ("regime shift of the North Atlantic subpolar gyre toward a persistent weak phase") rather than the first part of the clause ("eruption triggered"). Therefore, there is no contradiction. As our discussion makes clear, we do not believe our results have resolved the issue of whether the 1600 Huaynaputina eruption was the trigger for persistent winter cooling. In fact, we have found that winter cooling and expanded sea ice preceded the eruption, thus reducing the probability that an eruption triggered an SPG shift in the real world.
We have revised this portion of the conclusion to read: "This study has examined high-resolution proxies and historical observations to investigate whether the 1600 Huaynaputina eruption triggered persistent cooling in the North Atlantic region by initiating an SPG slowdown mechanism identified in previous modeling studies. Although the high-resolution reconstructions and historical observations of summer and winter temperature, wind direction, and sea-ice extent are consistent with such an eruption-induced mechanism, the results are inconclusive, particularly since reconstructions and observations indicate that the onset of winter cooling and increased sea ice may have preceded the Huaynaputina eruption."

An important information that was also never mentioned in the manuscript is that a number of volcanic reconstructions is available that have already been used for simulating the impact of volcanic eruptions on climate (e.g. Crowley and Unterman (2013); Gao et al. (2008), Toohey et al. (2016)). Especially the strength of larger tropical eruptions can vary up to a factor of two within the change in aerosol optical depth (AOD), the most important radiative physical moment in the stratosphere in the context of explosive volcanic eruptions. This should and could be taken into account by including a 2nd ESM simulation (e.g. the CCSM4 CMIP6 model used the Gao et al. 2008 data set in contrast to the Toohey et al. 2016 volcanic data set used in the present simulation). Integrating a 2nd set of simulations would help to better assess the impact/change of the SPG on the climate in Europe in the different Earth System Models.

Details on the volcanic forcing were already provided in Moreno-Chamarro et al. (2017a, 2017b) and in more detail in Jungclaus et al. (2014). Re-testing the SPG mechanism and its

sensitivity to volcanic forcing in additional climate models would be beyond the scope of the paper.

*Statistical Tests*
The general setup of the manuscript would greatly benefit by implementing a statistical test scheme with a clear formulation of a Null hypotheses that is falsified by an appropriate statistical test. Especially in the virtual world of the Earth System model this could be (quite easily) achieved. An option is for instance to design a test in the context of a bootstrap method: The null hypothesis is that the SPG has no influence on European temperatures. The nominal level can be set even to a two-sided test with 5 % . The test can now formally be carried out using sub-samplings of the different trajectories of the SPG in terms of block bootstrap by using control simulations. The test should be applied to the canonical pattern between the state of the SPG and European temperatures. If the sub-sampling leads not to statistically significant negative deviations of European temperatures in the presence of a shift in SPG, then the null hypothesis cannot be rejected. Eventually, these tests should be carried out for simulations with and without volcanic forcing.

It is not our hypothesis in this study that the SPG does/doesn't influence European temperatures.  Our question is: If the SPG-shift mechanism examined in previous studies did occur, could it have been triggered by the 1600 Huaynaputina eruption?
We attempted to identify precise initial conditions for the onset of the SPG-shift mechanism but were unable to find such an initial seed.  It is precisely because we were unable to determine a precise set of initial conditions required for the SPG shift (such as a state of the NAO) that we focused our study on identifying the precise timing of climatic and environmental changes associated with the SPG shift.  Although timing alone could not definitely determine whether or not the eruption triggered the SPG shift, it could add strong weight to either inference.  If climatic and environmental changes associated with the SPG shift had begun several years after the eruption, as found in the simulations, then we could have concluded that the eruption trigger was more probable *a posteriori*.  In fact, we found that those changes -- including increased sea ice and winter cooling -- commenced before the eruption.  This finding does not eliminate the possibility of an eruption trigger, since the pre-eruption changes could have arisen due to internal variability or other unidentified forcings.  However, it makes an eruption trigger less probable *a posteriori*.

We have expanded the relevant portion of discussion section 5.1 to read: "Our results, although consistent with the hypothesis that the Huaynaputina eruption triggered an SPG shift, do not provide additional support for this hypothesis. As discussed in section 2, simulations with an SPG shift showed characteristic short-term summer cooling and long-term winter anomalies, including persistent cooling and sea-ice expansion. The large degree of short-term summer cooling in the NVOLC v2 reconstruction is similar to the mean of simulations with an SPG shift. Nevertheless, the spatial pattern of the anomaly differs between the reconstruction and those simulations. There are several possible sources for this discrepancy.  The NVOLC v2 reconstruction has weaker spatial coverage in southeastern Europe, where some of the strongest differences appear (Guillet et al., 2017). The climate model necessarily simplifies climatic processes. Most importantly, this study has considered a relatively small ensemble size (10 realizations with volcanic forcing and 10 without) within a single model, as well as a single reconstructed history of volcanic forcing. Beyond limitations due to the specificity of the chosen model and forcing, the ensemble size seems to be insufficient to encompass the range of possible climate responses to the Huaynaputina

eruption that stems from their dependency on the initial state of the climate system at the time of the eruption.

Regarding winter anomalies, historical observations of North Sea wind direction indicated an unusually high frequency of winter northerly to easterly winds throughout the decade before and after 1600, especially in 1601. However, the series was too short and incomplete to demonstrate a long-term shift following the eruption. The Baltic harbor dates also indicated a short-term anomaly in 1601 as well as a multi-decadal period of colder, icier conditions. However, the change-point analysis indicates that this transition to colder icier conditions began before the Huaynaputina eruption, rather than several years following the eruption, as would have been expected with an eruption-triggered SPG shift. The historical observations of North Atlantic sea ice, taken in conjunction with previous paleoclimate studies discussed in section 3.4, appear to confirm this timing. In general, observers found icier conditions starting ca.1600, rather than another decade following the Huaynaputina eruption. Therefore, the reconstructions are consistent with at least two climatic scenarios, each found in different sets of simulations. In the first scenario, the 1600 Huaynaputina eruption triggered the SPG slowdown, but the shift to colder and icier conditions in Northern Europe and the North Atlantic had already begun by 1600 due to intrinsic climate variability or a different externally forced mechanism. This latter possibility stems from the fact that the 1600 Huaynaputina eruption was only the final eruption of a volcanic cluster that started in 1585. In the second scenario, the SPG slowdown commenced by 1600 without any role for the Huaynaputina eruption. Furthermore, it is possible that none of the simulations has reproduced the mechanism for persistent cooling that operated in the real world. Thus, our examination of high-resolution proxies and observations neither confirms nor disproves an eruption trigger for the previously proposed SPG-shift mechanism but it does make such a trigger less probable a posteriori."

*Physical mechanisms*
The authors mention a couple of (important) physical mechanisms that might support their hypotheses.
A first example relates to the NAO: at several places in the manuscript (cf. l. 69; l. 297) the authors mention the North Atlantic Oscillation as physical mechanisms explaining part of the temperature variability and being important also in the context of volcanic eruptions citing different authors. I wonder why the authors do not briefly explain the main mechanism suggested for the NAO in the first winter after volcanic eruptions (so called mid-Winter warming in Europe because of a positive state of the NAO, Kirchner, 1999; Zambri et al., 2017;). What mechanism is giving rise to such a response ? How robust is such kind of response and what effect does it exert on the winter temperatures in Europe ?

We thank the reviewer for drawing this issue to our attention. As discussed in Hernández et al. 2020, the precise NAO values are uncertain but both the Ortega et al. 2015 and Trouet et al. 2009 studies indicate roughly average NAO index values in the 1590s, declining in the decade following the 1600 eruption. Thus, the state of the NAO would not appear to be a strong explanation for the cooling before the eruption; nor is there evidence for an NAO+ response following the Huaynaputina eruption, unlike some other tropical eruptions. As recent studies indicate (Bittner et al., 2016; Coupe and Robock, 2021) a post-eruption NAO+ response with Eurasian winter warming appears to be contingent on tropospheric conditions at the time of the eruption rather than an automatic response to stratospheric aerosols.

We have added the following to the manuscript at line 169: "Nor did we find that the onset of the SPG shift depended on a particular state of the NAO. Neither simulations with an SPG

nor those without display consistent or anomalous high or low NAO index values in the decades before or after 1600, as shown in Figure 2. Furthermore, different reconstructions indicate different NAO index values during this period (**Figure 2** and Hernandez et al., 2020), but neither the reconstructions nor simulations display a positive NAO anomaly after 1600 such as those identified following other large tropical eruptions (e.g., Christiansen, 2008). As recent studies indicate (Bittner et al., 2016; Coupe and Robock, 2021), a post-eruption positive NAO response with Eurasian winter warming appears to be contingent on tropospheric conditions at the time of the eruption rather than a dynamical response to stratospheric aerosols alone." The new figure 2 illustrates NAO index reconstructions and values in simulations with and without volcanic forcing and SPG shift.

It is precisely because we were unable to determine a precise set of initial conditions required for the SPG shift (such as a state of the NAO) that we focused our study on identifying the precise timing of climatic and environmental changes associated with the SPG shift. Although timing alone could not definitely determine whether or not the eruption triggered the SPG shift, it could add strong weight to either inference. If climatic and environmental changes associated with the SPG shift had begun several years after the eruption, as found in the simulations, then we could have concluded that the eruption trigger was more probable *a posteriori*. In fact, we found that those changes -- including increased sea ice and winter cooling -- commenced before the eruption. This finding does not eliminate the possibility of an eruption trigger, since the pre-eruption changes could have arisen due to internal variability or other unidentified forcings. However, it makes an eruption trigger less probable *a posteriori*.

Sources:

Bittner M, Schmidt H, Timmreck C, Sienz F. Using a large ensemble of simulations to assess the Northern Hemisphere stratospheric dynamical response to tropical volcanic eruptions and its uncertainty. Geophys Res Lett. 2016;43(17):9324–32
Coupe, J, and Robock, A.: The influence of stratospheric soot and sulfate aerosols on the Northern Hemisphere wintertime atmospheric circulation. J. Geophys. Res. Atmos., 126, e2020JD034513, doi:10.1029/2020JD034513, 2021
Hernández, Armand, Celia Martin-Puertas, Paola Moffa-Sánchez, Eduardo Moreno-Chamarro, Pablo Ortega, Simon Blockley, Kim M. Cobb, et al. "Modes of Climate Variability: Synthesis and Review of Proxy-Based Reconstructions through the Holocene." Earth-Science Reviews 209 (2020): 103286. https://doi.org/10.1016/j.earscirev.2020.103286.

A second mechanism mentioned in this context relates to changes in blocking frequencies that are believed to be larger in the aftermath of volcanic eruptions and/or are an important mechanism explaining cold and very cold winters over (western) Europe (l. 66 ff). The authors argue that the blocking is independent to changes in the NAO. This is a bit surprising, because the NAO is the leading mode in Europe's winter variability and changes in the blocking should also effect the state of the NAO.

The increase in blocking frequency during the Little Ice Age was not associated with a persistent change in NAO phase in the simulations (Moreno-Chamarro et al., 2017b). Although a link between the NAO and blocking frequency has been established from observations, it usually refers to interannual variability. The time scales considered in the model were much longer (multidecadal to centennial) for which the NAO-blocking link

might work differently. Furthermore, the model might misrepresent their link, but this has not been explored and such assessments would be beyond the scope of the paper.

A third mechanisms relates to the role of sea ice (l 220 ff.). First, also sea ice concentrations can show a spatially heterogeneous pattern, especially when the entire North Atlantic region including Greenland is taken into account. In this context changes in the NAO can lead to dipole patterns with anomalous high sea ice around Greenland, and low sea ice over western Europe and vice versa. If this is not the case at least it should be motivated which canonical Circulation-sea ice patterns could lead to a spatially homogeneous response and/or whether direct radiative changes caused by volcanic eruptions could compensate or offset dynamically induced dipole patterns.

The dipole response of sea ice to NAO found in observations operates on annual to multi-annual time scales (Bader et al., 2011). The sea ice response follows a multidecadal to centennial reduction in the SPG and the related oceanic heat transport in the model, as explored in Moreno Chamarro et al. (2017a,b). The two mechanisms can be complementary since they operate on different time scales.

Reference:
Bader, J., Mesquita, M. D., Hodges, K. I., Keenlyside, N., Østerhus, S., and Miles, M.: A review on Northern Hemisphere sea-ice, storminess and the North Atlantic Oscillation: Observations and projected changes, Atmos. Res., 101, 809–834, https://doi.org/10.1016/j.atmosres.2011.04.007, 2011

A last mechanism I would like to mention here relates to the direct vs. indirect effects of volcanic eruptions on climate:
*The summer cooling is, by contrast, absent in the no-shift ensemble, which comprises mainly simulations without volcanic forcing (8 out of 12), the two ensembles show minor differences in oceanic variables such as the barotropic stream function and winter sea-surface temperature in the North Atlantic, which are weakly impacted by the volcanic forcing in the short term. Larger differences in these variables between the ensembles emerge over the following decades, particularly after the 1610s, in association with the SPG slowdown, as shown in Figure 3.[ l 162 ff. ]*
Here the authors even state that the summer cooling is absent in those simulations without volcanic forcing. Therefore the question remains as to whether a change in SPG is really necessary to initiate the sustained cooling. Also, a more objective formulation how the shift is quantified would be necessary to test if the deviation from the mean state is large enough to speak of a regime shift.

We intended to communicate that the summer cooling associated with the direct radiative effect from the volcanic forcing is absent in those simulations without volcanic forcing. Therefore, there is no contradiction here.
We have revised this portion of the text to read: "This short-term cooling mainly reflects the direct radiative response to the volcanic forcing, since 6 out of the 8 simulations that produce the SPG shift are those with volcanic forcing.  In the short term, simulations with an SPG shift also show minor differences in oceanic variables such as the barotropic stream function and winter sea-surface temperature in the North Atlantic. Larger differences between ensembles with and without an SPG shift-emerge over the following decades, particularly after the 1610s, as shown in **Figure 4**."

An SPG weakening is directly related to a multi-decadal to centennial cooling of the subpolar North Atlantic (Moreno-Chamarro et al., 2017a,b). However, we address a different question in this paragraph: whether the shift and no-shift ensembles show different short-term responses after the Huaynaputina eruption that can later result in an SPG shift.

*Synthesis with proxy and historical reconstructions*
In its present form the different chapters are not integrated into a consistent manner. Although this step is usually the most demanding, authors should at least indicate that their approach in comparing reconstructions with the world of the climate model is not a state-of-the art approach. For instance, forward models would be at hand to directly simulate respective proxies. In its present form the authors just use the information directly from the climate model without any further (advanced) processing. This represents an additional source of uncertainty in their conceptual framework.

As previously explained, our study does not aim to develop a new climate mechanism from paleo data and test it in the model world (or vice versa). Instead, we draw on new high-resolution proxies and historical observations to examine whether a previously derived and tested mechanism could have been triggered by an eruption.

Ours is one of the first studies to compare simulations with data from historical climatology (e.g., historical written descriptions and proxies from human activities) in this manner. Thus, the question of whether it is "state of the art" is a complicated one. On the one hand, these historical climatology data lack the spatial coverage, continuity, and homogeneity found in many paleoclimate reconstructions, and thus they are often less suited to deriving and testing climate mechanisms. On the other hand, they provide new kinds of precise, localized information unavailable in paleoclimate reconstructions and may therefore have unique and valuable applications in model-data comparison studies. In this case, our application of historical climatology concerned the timing of a possible mechanism to determine whether it could have been triggered by an eruption. Although our results turned out to be inconclusive, we anticipate they will guide future studies, as explained in lines 454-460.

The use of forward modeling for historical climatology is new and rare compared to that for paleoclimatology, since it presents the additional challenge of modeling what would be observed and recorded by a human observer. Nevertheless, it is feasible and may be a good plan for a future study study; therefore, we have added the following to our discussion section at line 450: "Moreover, forward modeling might be used to directly simulate additional proxies and conditions present to contemporary observers in order to provide stronger tests of an eruption trigger and SPG shift using historical climatology data. However, such modeling may have to take into account not only physical processes but human processes of observation, recording, and transmission."

**RC 5 with responses**

The study investigates the likelihood of a slowdown in the SPG around the onset of the LIA being linked to the 1600 Huaynaputina volcanic eruption. In order to resolve this issue, the authors attempt to integrate evidence from model-based simulations of past climate conditions with proxy-based paleoclimatic reconstructions and historical records. Despite the inconclusive results, the study highlights both the advantages of adopting an interdisciplinary approach as well as the challenges and limitations of bringing together and interpreting various sources of information.
**General comments:**

In my opinion, the multi-disciplinary nature of this study represents a considerable strength of this work. In general, the manuscript is well written and the findings are presented in a clear and logical manner. The evidence is interpreted objectively and the authors clearly acknowledge the limits of the analysis. From the presented results, conclusions are drawn to the extent that the simulation, reconstruction and limited observational data from the period allow. However, the unconventional structure is rather confusing since the introduction, methods description and some of the results are all blended together, and this also makes it somewhat difficult to distinguish for example what was done in previous studies and what represents original analysis. The authors should therefore seriously consider whether restructuring the manuscript in a more conventional format would be beneficial.

We would like to thank the reviewer for these generous comments on the manuscript. For the most part, the manuscript adopts a conventional structure: Introduction, Methods, Results, Discussion, and Conclusion. The only exception is an additional section (section 2) explaining the modeling and SPG-shift mechanism in the previous 2017 Moreno-Chamarro et al. studies. This additional section prevented the introductory section from becoming overly long and difficult to follow. It also enabled us to explain the previous modeling for a wider audience, including paleoclimatologists and historical climatologists, in keeping with the interdisciplinary scope of the article.
We have made several clarifications to the abstract and sections 1-3, including the following: The beginning of section 2 now reads: "This section reviews the previous modeling results that established the SPG mechanism for persistent cooling and its consistency with previous paleoclimate reconstructions, as well as the challenges in determining whether or not this mechanism was triggered by the 1600 Huaynaputina eruption."
The beginning of section 3 now reads: "This section presents the new high-resolution paleoclimate proxies and historical observations of climate and environmental conditions selected to determine whether these were consistent or not with a Huaynaputina eruption trigger for the previously identified SPG mechanism for persistent cooling."

Currently, a large part of the discussion is dedicated to discussing the historical / societal impacts of cold conditions at the end of the 16th and during the early 17th century. Greater focus on integrating and discussing the results of the modeling, proxy and historical datasets in more detail would be helpful.

Another important point is recognizing and acknowledging discrepancies between model-based simulations with proxy-based reconstructions, which has consequences for understanding uncertainty and the overall reliability of these data sources. This issue is highlighted for example by Figure 6, which shows poor spatial agreement between modeled and reconstructed temperatures. Model simulations are often associated with high uncertainty particularly in relation to post-volcanic cooling and, for example, over-estimation of the magnitude of post-volcanic cooling by some models has been known to occur (e.g. Chylek *et al.*, 2020; Hartl-Meier *et al.*, 2017). Better understanding of some of the shortcomings of these datasets and limitations in their utility within the context of this study could be achieved by exploring a broader set of model simulations or model types to help disentangle the possible influence of model bias and a more detailed examination of the proxy-based temperature reconstructions would also be helpful in this regard.

We have substantially revised and expanded section 5.1 in light of these comments. Please see the new text beginning on line 407.

It is also necessary recognize the potential importance of background climate conditions in modulating the (cooling) response of the North Atlantic to large volcanic eruptions based on the state of the climate system. In relation to this point, the role of internal variability and specifically the potential role of the North Atlantic Oscillation (NAO) in the initiation of SPG weakening and cooler conditions in the north Atlantic sector remains a subject of debate (e.g. Trouet *et al.*, 2009; Lehner *et al.*, 2012). For this reason, some type of examination and discussion of the modes of atmospheric variability in the north Atlantic within this context would be helpful.

We thank the reviewer for drawing this issue to our attention. The role of the background state in the onset of the SPG slowdown was previously discussed in the 2017 Moreno-Chamarro et al. studies. As discussed in Hernández et al. 2020, the precise NAO values are uncertain but both the Ortega et al. 2015 and Trouet et al. 2009 studies indicate roughly average NAO index values in the 1590s, declining in the decade following the 1600 eruption. Thus, the state of the NAO would not appear to be a strong explanation for the cooling before the eruption; nor is there evidence for an NAO+ response following the Huaynaputina eruption, unlike some other tropical eruptions. As recent studies indicate (Bittner et al., 2016; Coupe and Robock, 2021) a post-eruption NAO+ response with Eurasian winter warming appears to be contingent on tropospheric conditions at the time of the eruption rather than an automatic response to stratospheric aerosols.

We have added the following to the manuscript at line 169: "Nor did we find that the onset of the SPG shift depended on a particular state of the NAO. Neither simulations with an SPG nor those without display consistent or anomalous high or low NAO index values in the decades before or after 1600, as shown in Figure 2. Furthermore, different reconstructions indicate different NAO index values during this period (**Figure 2** and Hernandez et al., 2020), but neither the reconstructions nor simulations display a positive NAO anomaly after 1600 such as those identified following other large tropical eruptions (e.g., Christiansen, 2008). As recent studies indicate (Bittner et al., 2016; Coupe and Robock, 2021), a post-eruption positive NAO response with Eurasian winter warming appears to be contingent on tropospheric conditions at the time of the eruption rather than a dynamical response to stratospheric aerosols alone." The new figure 2 illustrates NAO index reconstructions and values in simulations with and without volcanic forcing and SPG shift.

Sources:

Bittner M, Schmidt H, Timmreck C, Sienz F. Using a large ensemble of simulations to assess the Northern Hemisphere stratospheric dynamical response to tropical volcanic eruptions and its uncertainty. Geophys Res Lett. 2016;43(17):9324–32
Coupe, J, and Robock, A.: The influence of stratospheric soot and sulfate aerosols on the Northern Hemisphere wintertime atmospheric circulation. J. Geophys. Res. Atmos., 126, e2020JD034513, doi:10.1029/2020JD034513, 2021
Hernández, Armand, Celia Martin-Puertas, Paola Moffa-Sánchez, Eduardo Moreno-Chamarro, Pablo Ortega, Simon Blockley, Kim M. Cobb, et al. "Modes of Climate Variability: Synthesis and Review of Proxy-Based Reconstructions through the Holocene." Earth-Science Reviews 209 (2020): 103286. https://doi.org/10.1016/j.earscirev.2020.103286.

One obvious limitation is that most of the presented evidence for the SPG shift is either indirect / circumstantial or entirely model-based. Although the study provides a compelling narrative characterizing anomalously cold conditions in the early 17th century, a certain leap of faith is currently required to link an SPG mode shift to these changes. In any case, more

information would be required to clarify the relationship between the eruption, short-term and long-term cooling and how these events and changes relate to the state of the SPG. Ultimately, there are limits to the answers that modeling can provide and additional more direct proxy data would likely be required to better understand the dynamics of oceanic circulation and atmospheric dynamics during this period to more precisely pin down the timing, duration and extent of the purported SPG slowdown. Perhaps then it would be possible to confirm or refute the attribution of the observed longer-term cooling in the early 17$^{th}$C, and by extension the initiation of an SPG slowdown, to a volcanic trigger.

The evidence for the SPG shift was provided in the 2017 Moreno-Chamarro et al. studies. This included testing against long-term mainly decadal- to multi-decadal-scale paleoclimate reconstructions. Those reconstructions were insufficient to determine whether the Huaynaputina eruption could have been the trigger for an SPG shift. The new data from high-resolution proxies and historical observations examined in this study enables more precise specification of conditions ca.1600, which raised the possibility of examining whether the previously proposed SPG shift could have been triggered by the Huaynaputina eruption. Additional paleoclimate reconstructions may help determine the possible duration, degree, and extent of an SPG shift; and those additional inferences might help determine whether an eruption triggered an SPG shift in the first place. Nevertheless, such an investigation would be beyond the scope of the current study. The simulations do not currently indicate different types of SPG shifts of different timing, duration, and extent, with some types always triggered by an eruption and others not. Moreover, there is a trade-off between the precise, localized, diverse information provided by historical climatology and the more long-term, continuous, homogenous information provided by paleoclimate reconstructions. Thus, the reconstructions previously used in the 2017 Moreno-Chamarro et al. studies to test the presence of an SPG shift in the real world were less suited for determining whether that shift was triggered by an eruption; while the high-resolution proxies and historical observations used in this study to test for an eruption trigger would be less appropriate for examining the duration or extent of an SPG shift.

**Specific comments:**
L63-72: While it may perhaps be possible for such changes to occur without invoking substantial changes to atmospheric dynamics in the North Atlantic, the background state of the atmosphere, internal variability and the role of the NAO cannot be discounted *a priori*, particularly as these factors may act to modulate the response of the climate system to a large volcanic event.

(Please see the above response to concerns about the NAO state at the time of the eruption.)

L89: The phrase 'possibilities for adaptation' seems a bit vague and it is not clear what this refers to. Please specify / clarify this point.

We have changed the phrase "nature of societal vulnerabilities and possibilities for adaptation" to "which activities and institutions were vulnerable, and how people could adapt them to changing climatic and environmental conditions".

Figure 2: For easier interpretation of the figure, it may be clearer to also state in the panel sub-headings that the plots are showing temp. / Sv. anomalies.

We thank the reviewer for the suggestion and have corrected the figure.

L233: It is not clear whether this implies that only a 30-yr segment length was used or a range of segment lengths (30-yr+) was examined. If it is the former case, please remove 'minimum' to avoid confusion. Otherwise, please specify the range of segment lengths utilized.

The segment length (minimum number of observations between the changes) in the change point analysis is ≥ 30-years. We have revised the line to: "we looked for changepoints in the Tallinn harbor ice breakup timing data for any year within any segment length of ≥30 years in the period 1550-1675"

Figure 5: Please specify in the figure caption what the purple dots in top-left plot represent.

The purple dots in top-left plot of Figure 5 are the tree-ring width and maximum latewood density sites that have been used for the spatial reconstruction. We have revised the figure caption to include this information.

L203-210: What was the size of the reconstructed grid cells? Which instrumental dataset was used for calibration? How were the chronologies merged and how was the reconstruction performed (e.g. PCA, nesting), etc.? In general, more detailed information about the development of the spatial reconstruction is needed here (or at least in supplementary materials).

The size of reconstructed grid cells is 5x5° lat/long. The instrumental data used as target field for the reconstruction are May to August monthly temperature anomalies wrt the 1961-1990 period extracted from the HadCruT4 (Cowtan and Way, 2014). The spatial reconstruction was developed using a point-by-point regression (Cook et al., 1999) which accounts for the spatial distribution and relationship of the proxy predictor network to the target field. We have expanded section 3.1 to include all this information.

References:
Cook, Edward R., David M. Meko, David W. Stahle, and Malcolm K. Cleaveland. "Drought Reconstructions for the Continental United States." Journal of Climate 12 (1999): 1145–62. https://doi.org/10.1175/1520-0442(1999)012<1145:DRFTCU>2.0.CO
Cowtan, Kevin, and Robert G. Way. "Coverage Bias in the HadCRUT4 Temperature Series and Its Impact on Recent Temperature Trends." Quarterly Journal of the Royal Meteorological Society 140, no. 683 (2014): 1935–44. https://doi.org/10.1002/qj.2297.

Figure 6: How does the NVOLC reconstruction compare with N-TREND (and model output) over the investigated period? Currently, only NVOLC is compared to model output, whereas N-TREND is only used for illustration and is not compared to NVOLC or the modeled temperatures. The highly anomalous cooling in SE Europe in the NVOLC reconstruction (Fig. 6a) is rather suspicious and I wonder how robust this feature is. According to Supplementary Figure S3 in Guillet et al. (2017), most of northern Europe and parts of western / southwest Europe calibrate well, whereas calibration / verification statistics are very weak for NW, central and especially eastern and SW Europe. Consider that poor spatial representation of reconstructed temperatures may cause disagreement with modeled temperatures in some areas. Likewise, specific limitations of the model may also lead to disagreement. Such considerations should be acknowledged and discussed.

We have added the following discussion regarding discrepancies between the simulations and NVOLC reconstruction starting at line 409: "The large degree of short-term summer cooling in the NVOLC v2 reconstruction is similar to the mean of simulations with an SPG shift. Nevertheless, the spatial pattern of the anomaly differs between the reconstruction and those simulations. There are several possible sources for this discrepancy. The NVOLC v2 reconstruction has weaker spatial coverage in southeastern Europe, where some of the strongest differences appear (Guillet et al., 2017). The climate model necessarily simplifies climatic processes. Most importantly, this study has considered a relatively small ensemble size (10 realizations with volcanic forcing and 10 without) within a single model, as well as a single reconstructed history of volcanic forcing. Beyond limitations due to the specificity of the chosen model and forcing, the ensemble size seems to be insufficient to encompass the range of possible climate responses to the Huaynaputina eruption that stems from their dependency on the initial state of the climate system at the time of the eruption."
Comparison with the N-TREND reconstruction may create confusion, however, since N-TREND includes many more TRW chronologies (with the problem of memory in series) than the NVOLC dataset, which was created with the goal to detect and quantify volcanic cooling and thus includes temperature-sensitive MXD records.

L286: Why is the NVOLC v2 reconstruction shifted by +0.5 K?

The aim of the figure is to compare the forcing generated by the 1600 eruption in the simulations and reconstruction rather than absolute temperatures.

L350: I suggest that a more appropriate term to use in this context would be 'support' rather than 'appear to confirm'.

We thank the reviewer for the suggestion and have revised the sentence accordingly.

L368-370: So, considering the timing, might this in fact suggest that the Huaynaputina eruption is rather unlikely to be the cause of the SPG slowdown?

It remains difficult to say. As we explain in the next paragraph of the manuscript, the results would be consistent with a scenario in which the eruption did trigger an SPG shift but colder conditions had already started during the 1590s due to internal variability or a different forcing. Therefore, we could say that the results do *not confirm* a scenario in which the 1600 eruption triggered an SPG shift, but we couldn't say that the results *contradict* such a scenario. In terms of inference, our findings should reduce posterior estimates of the probability for an eruption trigger (per Bayes' theorem) since the likelihood of our reconstruction data given an eruption trigger is lower than the likelihood of getting that data regardless of the eruption trigger. Whether the eruption trigger hypothesis is *a posteriori* improbable -- i.e., $p(h|d) < 0.5$ -- would depend on one's prior probability for the hypothesis. We have added the following sentence to the discussion section (starting line 433) for clarification: "Thus, our examination of high-resolution proxies and observations neither confirms nor disproves an eruption trigger for the previously proposed SPG-shift mechanism but it does make such a trigger less probable a posteriori."

L371-385: Another possibility could be that a pronounced shorter-term cooling impact of the Huaynaputina eruption was 'superimposed' on the longer-term cooling trend, which may have been initiated prior to 1600 (either in response to the cluster of late-16[th] century volcanic eruptions or otherwise). Evidence for volcanic-induced short-term cooling is on

firmer ground as the results are consistent with this type of response to the eruption (Fig. 6c) and this is also consistent with the duration and magnitude of inferred NH cooling responses to large (tropical) eruptions more generally based on proxy reconstructions (e.g. Esper et al. 2015) and modeling of surface air temperature. In contrast, the mechanism for initiating longer-term cooling / SPG slowdown and attribution of such changes to a particular volcanic event is highly uncertain and rather problematic.

We thank the reviewer for this suggestion. It is, of course, possible that none of the simulations have captured the mechanism for cooling found in the real world, and we have noted this possibility in the revised discussion section 5.1 (line 432). However, as previously discussed in the 2017 studies by Moreno-Chamarro et al., 8 of the 20 simulations (6/10 with volcanic forcing and 2/10 without) reproduced an SPG shift as well as long-term winter cooling and increased sea ice extent found in previous reconstructions. Thus, we have chosen to investigate this mechanism more closely and to assess different scenarios indicated by those simulations.

L376-377: One could argue that it is uncertain whether this issue could be definitively resolved through modelling alone.

We have revised the sentence to read "First, *further comparison between high-resolution reconstructions and* a larger ensemble of climate simulations could improve…"

L395-397: I would recommend reformulating this sentence considering that, based on extensive paleoclimatic evidence, the occurrence of cool (wet) summers during this period is actually not in question. Therefore, rather than 'confirming' this, it would be more appropriate to state that this study provides further support and a broader context for such conditions at that time.

We thank the reader for this suggestion and have revised the sentence accordingly.

**Minor / technical comments:**
L95: consider '… activation (or lack thereof) …'

L186/380/384: 'an SPG' rather than 'a SPG'

Fig.4 legend / L233 / (L303): 'ice break-up timing data' instead of 'date data'

L223: consider 'obtained' instead of 'gained'

L238: consider 'recorded' rather than 'left'

L239: 'latter' rather than 'later'?

L243: Suggested wording adjustment: 'These observations were recorded in areas with flat terrain …'

L266: 'did or did not turn back' or alternatively 'ships could pass or were forced to turn back'

L267: 'turn back during a voyage' or perhaps 'terminate a voyage'?

L268: 'cold conditions' or 'cold temperatures' / 'dangerous sailing conditions' or 'danger posed by sailing conditions'

L280: Change 'NTREND' to 'N-TREND'. Also, something is missing here - consider: '… in each year over the 1601-1609 period …'?

L287: should the range be 1593-1650 instead of 1593-1640?

L290: Consider: 'The analysis in Figure 7 indicating the ice …'

L293: 'can occur' rather than 'can happen'

L303: please change 'wrt' to 'w.r.t.'

L308: 'detected in' rather than 'detected at'?

L362: 'as a Possible Trigger'?

L372: remove 'has'?

L375: 'any role of'?

We thank the review for reading the manuscript closely. We have corrected the typos and clarified the phrasing in the lines indicated above.

---

## Author Response (AR2)

**Response to reviewer comments, minor revisions**

General:

The manuscript has been improved with respect to the first version, especially concerning a more focused statement of the general setup and potential weaknesses of the model/proxy comparisons and the fact that model results are entirely based on earlier publications of Moreno-Chamarro et al., (2017a, 2017b). Therefore I think in its present form it is conceptually better framed, also taking into account the more general readership of paleoclimatologists.

We'd like to thank the reviewer the comments here and on the previous draft, which have improved the clarity of this paper.

Below are a few comments from the replies that should be addressed in the final version of the manuscript.

Specific:

As outlined, specific concerns were not been addressed because the basic physical and statistical setup was completely based on earlier studies of Moreno-Chamarro et al. (2017a, 2017b), not being intended for additional investigations in the present manuscripts. Therefore most comments e.g. related to the setup of a more objective statistical testing scheme have not been implemented in the update of the manuscript.

That is correct. The study did not involve new modeling but instead used new paleoclimate and historical climatology data to assess the possibility of an eruption trigger for the previously hypothesized SPG slowdown mechanism (Moreno Chamarro et al., 2017a, 2017b).

The remaining comments are adequately addressed in the light of using the results (not the output) of the MPI-ESM-P simulations for (semi-) quantitative comparisons with the proxy data introduced and analysed in the context of links to potential SPG changes.

Introduction: I would like to suggest to include the following paragraph with according references formulated in the reply into the final version into the introduction of the study (e.g. line 89 ff):

Including other last millennium simulations would have blurred the discussion because it would have meant dealing with different models and model sensitivity to external forcings, different volcanic forcings, different background climate states at the time of the eruption, and different internal variability. The only model with a close enough setup is the CESM last millennium ensemble, which includes sensitivity simulations with different external forcing. However, it is not clear whether this model, the CESM-CAM5_CN, shows any sensitivity in the subpolar region to the volcanic forcing (Otto-Bliesner et al., 2016), although a newer model version shows cooling in the North Atlantic during the Little Ice Age associated with a SPG weakening (Zhong

et al., 2018). There is currently limited data for other CMIP6/PMIP4 last millennium simulations available at the ESGF nodes (and not for the MPI-ESM model).

We have included a modified version of this paragraph after line 105 and modified some of the surrounding text to explain the choice of climate model.

Beginning of section 2 and 3: A great improvement is that some weaknesses of the approach are explained in greater detail in the new version of the manuscript and also the focus is properly set to make the reader already aware on the core of the study.

l. 450: This paragraph also helps to address and motivate for outlook and follow-up studies (cf. Forward modeling)

l. 435: I don t understand what the a posteriori line of evidence means – what does this really mean if one could a posteriori say that something is more probable, i.e. having already the knowledge of the outcome ? Maybe the authors should re-formulate, e.g. that chances for a slowdown of the SPG after a volcanic eruption are slightly higher compared to a situation without a volcanic eruption ?

The a posteriori probability referred to our degree of believe in the hypothesized eruption trigger after analyzing our new data.  Since this wording was confusing, we have removed the phrase.